# Structures in multiple conformations reveal distinct transition metal and proton pathways in an Nramp transporter

Aaron T Bozzi[†], Christina M Zimanyi[†‡], John M Nicoludis, Brandon K Lee, Casey H Zhang, Rachelle Gaudet*

Department of Molecular and Cellular Biology, Harvard University, Cambridge, United States

**Abstract** Nramp family transporters—expressed in organisms from bacteria to humans—enable uptake of essential divalent transition metals via an alternating-access mechanism that also involves proton transport. We present high-resolution structures of *Deinococcus radiodurans* (Dra)Nramp in multiple conformations to provide a thorough description of the Nramp transport cycle by identifying the key intramolecular rearrangements and changes to the metal coordination sphere. Strikingly, while metal transport requires cycling from outward- to inward-open states, efficient proton transport still occurs in outward-locked (but not inward-locked) DraNramp. We propose a model in which metal and proton enter the transporter via the same external pathway to the binding site, but follow separate routes to the cytoplasm, which could facilitate the co-transport of two cationic species. Our results illustrate the flexibility of the LeuT fold to support a broad range of substrate transport and conformational change mechanisms.
DOI: https://doi.org/10.7554/eLife.41124.001

*For correspondence:
gaudet@mcb.harvard.edu

[†]These authors contributed equally to this work

Present address: [‡]New York Structural Biology Center, New York, United States

Competing interests: The authors declare that no competing interests exist.

## Introduction

The amino acid-polyamine-organocation (APC) superfamily of secondary transporters encompasses a broad range of evolutionarily-related proteins that transport diverse substrates including neurotransmitters, metabolites, and transition metals in organisms throughout the tree of life (*Vastermark et al., 2014*; *Wong et al., 2012*). In humans alone, the APC superfamily encompasses 11 subfamilies of distinct solute carrier proteins (*Perland and Fredriksson, 2017*). These transporters harness the energy stored in preexisting transmembrane ion gradients. The LeuT fold (*Yamashita et al., 2005*) is the core structural unit that undergoes conformational rearrangements necessary for alternating access-based transport in the APC superfamily. This fold consists of ten transmembrane (TM) segments, divided into two pseudosymmetric, interlocking five-TM repeats, although many members have additional TMs. Primary substrates bind in a pocket formed by non-helical regions of TM1 and TM6, close to the center of the membrane. Co-transported coupling ions—typically $Na^+$ and/or $H^+$—bind at the interface between two proposed domains (*Perez and Ziegler, 2013*; *Rudnick, 2013*): a 'bundle' formed by TMs 1, 2, 6, and 7; and a 'scaffold' or 'hash' domain comprising most or all of the remaining six TMs (*Forrest and Rudnick, 2009*). When all substrates are bound, conformational rearrangement closes an external vestibule between 'bundle' and 'scaffold' and opens an intracellular vestibule between the two domains to allow substrate release (*Boudker and Verdon, 2010*; *Forrest et al., 2011*; *Shi, 2013*). Despite the common fold, many APC members have little-to-no sequence identity, consistent with mechanistic divergences, including variance in the identity and stoichiometry of the coupled ions (*Ma et al., 2012*; *Shaffer et al., 2009*) and in which helices move the most to open and close the inner and outer gates (*Kazmier et al.,*

**eLife digest** Cells use transport proteins embedded in their membrane to acquire many of the nutrients they need to survive and grow. Different transport proteins transport different nutrients; for example, the Nramp transporters move transition metal ions across cell membranes. Nramps are found in a wide range of organisms. Bacteria use them to acquire the metals they need during the course of an infection, and humans rely on Nramps to absorb iron from food. Nramps can also transport hydrogen ions (known as protons).

Understanding how the structure of an Nramp transporter changes as it transports metal ions and protons can help researchers to understand how it works. These structures can be studied using a technique called X-ray crystallography, which captures snapshots of the proteins at different stages of their task.

Bozzi, Zimanyi et al. used X-ray crystallography to study the structures of an Nramp transporter from the bacterium *Deinococcus radiodurans*. The results reveal four of the shapes that the Nramp transporter takes on at different stages in its transport process, including the first structure to show an Nramp binding to a metal ion from the outside of the cell. Taken together, the structures suggest a new transport mechanism that has not been seen in previously studied transport proteins with similar structures. An unexpected feature of this mechanism is that Nramps transport metal ions and protons along different pathways.

Studying the transport mechanisms used by Nramp transporters will help researchers to understand how cells maintain appropriate levels of metal ions, an important component of human health. The mechanisms of relatively few transport proteins are understood at a structural level, yet many share common origins and have shared characteristics. Understanding how Nramps work could therefore help us to understand how wider classes of transporters work as well.
DOI: https://doi.org/10.7554/eLife.41124.002

*2017*; *Kazmier et al., 2014a*; *Kazmier et al., 2014b*; *Krishnamurthy and Gouaux, 2012*; *Malinauskaite et al., 2014*; *Ressl et al., 2009*; *Shimamura et al., 2010*; *Weyand et al., 2008*).

Natural resistance-associated macrophage proteins (Nramps) are APC-superfamily transition metal transporters that enable uptake of rare micronutrients such as $Mn^{2+}$ in plants and bacteria and $Fe^{2+}$ in animals (*Cellier, 2012*; *Courville et al., 2006*; *Nevo and Nelson, 2006*). Nramps bind and/or transport biologically-essential divalent metals such as $Mn^{2+}$, $Fe^{2+}$, $Co^{2+}$, $Ni^{2+}$, $Cu^{2+}$, $Zn^{2+}$—and toxic metals like $Cd^{2+}$, $Pb^{2+}$, and $Hg^{2+}$—but not the abundant alkaline earth metals $Mg^{2+}$ and $Ca^{2+}$ (*Bozzi et al., 2016a*; *Ehrnstorfer et al., 2014*). Metal uptake by Nramps is typically stimulated by acidic pH and accompanied by proton influx (*Chen et al., 1999*; *Ehrnstorfer et al., 2017*; *Gunshin et al., 1997*). However, many homologs also display considerable proton uniport—proton transport in the absence of added metal that suggests loose, if any, coupling between the two substrates (*Chen et al., 1999*; *Gunshin et al., 1997*; *Mackenzie et al., 2006*; *Nelson et al., 2002*; *Xu et al., 2004*). To date no studies have conclusively demonstrated that Nramp is in fact a thermodynamically coupled secondary transporter capable of harnessing a favorable gradient of metal or proton to power electrochemically-uphill transport of the other substrate.

Nramps have 11 or 12 TMs, the first ten forming a LeuT fold, as seen in structures of three bacterial Nramp homologs (*Bozzi et al., 2016b*; *Ehrnstorfer et al., 2014*; *Ehrnstorfer et al., 2017*), including our model system *Deinococcus radiodurans* (Dra)Nramp (*Bozzi et al., 2016b*). Conserved aspartate, asparagine, and methionine residues in TM1 and TM6 coordinate transition metal substrates as observed in an inward-open state (*Ehrnstorfer et al., 2014*), while only a metal-free outward-open state has been reported (*Ehrnstorfer et al., 2017*).

Here, we provide the first complementary structures of the same Nramp homolog in multiple conformations, including the first metal-bound outward-open Nramp structure, and a novel inward-occluded structure. These allow us to fully illustrate the transport cycle for DraNramp. We also show that metal transport requires the expected alternating access bulk conformational change, whereas proton transport can occur via a more channel-like mechanism in the outward-open state. Using the structures and accompanying biochemical data, we delineate separate conserved transport pathways for metal and proton substrates and provide a mechanistic model encompassing substrate

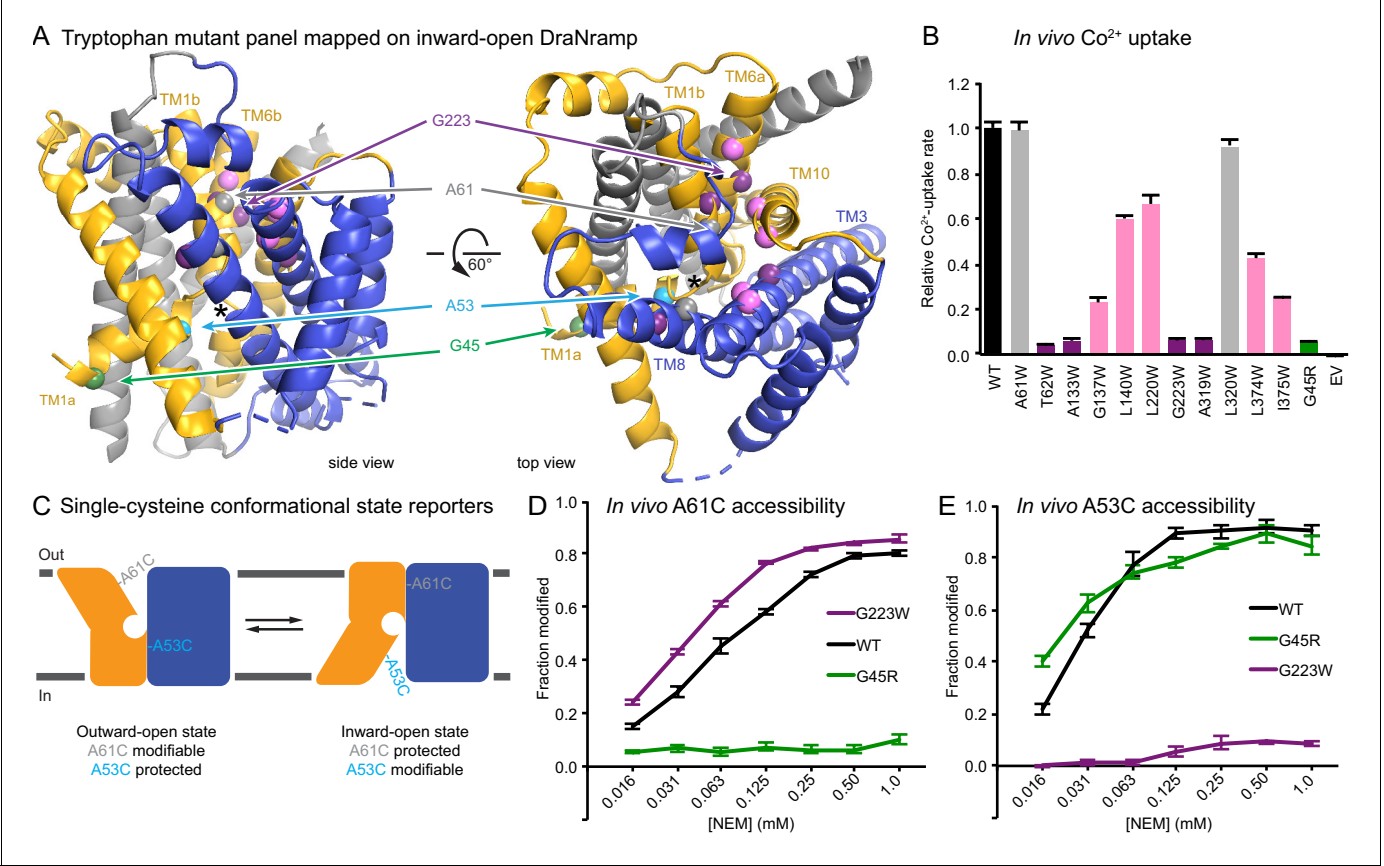

**Figure 1.** Design and validation of complementary conformationally-locked crystallization constructs. (**A**) Panel of 11 designed tryptophan mutants (pink, purple, and gray spheres; see (**B**)) and G45R disease-mutant mimic (green) mapped onto our initial inward-open DraNramp structure and color-coded by their effect on transport activity in (**B**). Single-cysteine reporters A53C (cyan) and A61C (gray) are indicated. TMs 1, 5, 6, and 10 are colored gold, TMs 3, 4, 8, and 9 blue, and TMs 2, 7, and 11 gray. * marks the approximate location of the metal-binding site. (**B**) Relative in vivo $Co^{2+}$ uptake rates for tryptophan mutant panel. Mutants colored gray, pink and purple did not affect, moderately decreased, or eliminated uptake, respectively. (**C**) Two complementary single-cysteine reporters enable assessment of the mutants' conformational state preferences. For simplicity, in schematics DraNramp is depicted as two gold and blue domains that reorient to accomplish alternating access. (**D**) G45R prevented modification of the A61C outward-reporter, and (**E**) G223W fully protected the A53C inward-reporter, indicating that complementary conformational locking was achieved. Both reporters fully labeled for the WT-like protein, likely because it cycles between both conformations during the assay. Data in B, D, and E are averages ± S.E.M. (n = 4). See also **Figure 1—figure supplement 1**.

DOI: https://doi.org/10.7554/eLife.41124.003

The following figure supplement is available for figure 1:

**Figure supplement 1.** Validation of conformationally-locked crystallization constructs.
DOI: https://doi.org/10.7554/eLife.41124.004

binding, release, and the conformational change process. We demonstrate novel modes of conformational rearrangement and ion shuttling in DraNramp compared to other LeuT-fold transporters, thus expanding the known repertoire of intramolecular dynamics and substrate transport mechanisms possible within this important protein family.

# Results

## Rational design and validation of conformationally-locked DraNramp constructs

A previously determined structure of a Fab-bound DraNramp in an inward-open conformation revealed the intracellular metal permeation pathway, or vestibule, between TMs 1a, 2, 5, 6b, 7, and 8 (*Bozzi et al., 2016b*). This structure was stabilized in an inward-open state by patches of mutations

to intracellular loops 4–5, 6–7, and 10–11, and we thus refer to it as the Patch mutant. To observe additional conformational states of a transport cycle in a single Nramp homolog at high resolution, we developed two complementary conformationally-locked constructs for crystallization. Adding steric bulk along TM1a—for example a G45R mutation, which mimics a human anemia-causing mutation of a conserved glycine (*Barrios et al., 2012*)—prevented the opening of the extracellular vestibule and eliminated metal transport, emphasizing the importance of the alternating-access mechanism to DraNramp function (*Bozzi et al., 2016b*). Based on these findings we pursued the G45R mutant as a new inward-locked crystallization construct.

To develop a complementary outward-locked DraNramp construct, we adapted an approach previously described for the lactose transporter LacY (*Kumar et al., 2014*; *Smirnova et al., 2013*). By mapping extensive cysteine accessibility data onto the inward-open structure, we identified the external vestibule between TMs 1b, 6a, 3, 8, and 10 (*Bozzi et al., 2016b*). We created a panel of 11 tryptophan point mutants lining this predicted external vestibule (*Figure 1A*) to destabilize the inward-open state. An outward-locking mutation should severely impair metal transport, and indeed several mutants had impaired in vivo $Co^{2+}$ uptake when expressed in *Escherichia coli* (*Figure 1B* and *Figure 1—figure supplement 1A*). We chose to pursue G223W—on TM6a one helical turn above the unwound metal-binding region—which like G45R eliminated $Co^{2+}$ and $Fe^{2+}$ metal transport (*Figure 1—figure supplement 1B*). Importantly, a tryptophan modeled in the inward-open DraNramp Patch mutant structure at position 223 clashes with the top of TM10. In contrast, in the recent outward-open structure of *Eremococcus coleocola* (Eco)Nramp (33% identity with DraNramp) the analogous glycine lines a wide aqueous channel with adequate room for tryptophan's bulk (*Ehrnstorfer et al., 2017*).

To further validate this G223W construct, we measured bulk solvent accessibility of two single-cysteine reporters: A61C on TM1b, which is exposed only in DraNramp's outward-open state (*Bozzi et al., 2016b*); and A53C on TM1a just below the metal-binding D56, a putative inward-open reporter based on comparing the Patch mutant and EcoNramp structures (*Figure 1C* and *Figure 1—figure supplement 1C*). WT-like DraNramp (with the indicated reporter cysteine and a C382S mutation to remove the lone endogenous cysteine) maintains a dynamic conformational equilibrium—even in the absence of added metal substrate—such that either reporter can be fully modified by the thiol-specific N-ethylmaleimide (NEM) at high concentrations (*Figure 1D–E*). G45R slightly increased A53C accessibility but fully protected A61C, indicating an outward-closed state that we will refer to as inward-locked based on these data and the structure described below. In contrast, G223W significantly increased A61C accessibility while fully protecting A53C, consistent with an outward-locked state (*Figure 1D–E* and *Figure 1—figure supplement 1D*). We have thus identified two complementary constructs that trap DraNramp in outward-locked (G223W) and inward-locked (G45R) states (*Figure 2—figure supplement 1A*). Using lipidic cubic phase (LCP) to mimic the hydrophobic membrane environment, we crystallized and determined the structures of G45R and G223W to resolutions of 3.0 and 2.4 Å, respectively, both significantly improved from our earlier DraNramp structure (3.94 Å) (*Table 1* and *Figure 2—figure supplements 1–2*). The new high-resolution structures also allowed us to re-refine our original structure, including correction of a sequence registry error in TM11.

## Structure of G45R DraNramp reveals inward-occluded state

Unexpectedly, the G45R structure is not in an inward-open conformation as seen previously with our Fab-bound Patch mutant (*Figure 2A*) but instead adopts an inward-occluded, metal-free state (*Figure 2B*) that may represent an intermediate conformation between inward-open and outward-open states in the DraNramp transport cycle (*Figure 2D*). As in the inward-open apo state, the external vestibule remains sealed, with TM1b and TM6a forming tight hydrophobic packing with the tops of TM3 and TM10, and most TMs undergo little apparent displacement (*Figure 2F–G*). The major exception is TM1a, which swings ~45° to partially seal the inward aqueous cavity in the G45R structure, a motion we previously showed to be essential to the transport cycle (*Bozzi et al., 2016b*). The intracellular ends of TM4 and TM5 also move slightly compared to their position in the inward-open state, further sealing the metal-binding site from the cytosol. Comparisons of the G45R and G223W structures indicate that, rather than preventing inward motion of TM1a as we had hypothesized (*Bozzi et al., 2016b*), the G45R mutation precludes TM4-TM5 from fully closing the inner gate, as any bulkier replacement for that absolutely-conserved glycine in our outward-open

**Table 1.** DraNramp data collection and refinement statistics.

| | Inward-open apo (Patch mutant + Fab) | Inward-occluded apo (G45R) | Outward-open Mn$^{2+}$-bound (G223W ΔN34) | Outward-open apo (G223W ΔN34) |
|---|---|---|---|---|
| PDB ID | 6D9W | 6C3I | 6BU5 | 6D91 |
| SBGrid Data Bank ID | 332, 333, 334 | 567 | 564 | 576 |
| **Data Collection** | | | | |
| Wavelength (Å) | 1.139 | 0.9793 | 0.9792 | 0.9792 |
| Resolution range (Å) | 46.47–3.94 (4.08–3.94) | 50.00–3.00 (3.05–3.00) | 39.19–2.40 (2.49–2.40) | 38.14–2.36 (2.44–2.36) |
| Space group | I222 | P2$_1$2$_1$2$_1$ | C121 | C121 |
| Unit cell (a, b, c) | 113.13, 132.08, 221.0 | 70.73, 97.85, 118.26 | 105.76, 80.39, 51.75 | 101.92, 75.51, 53.15 |
| Unit cell (α, β, γ) | 90.0, 90.0, 90.0 | 90.0, 90.0, 90.0 | 90.0, 94.72, 90.0 | 90.0, 98.07, 90.0 |
| Number of crystals | 3 | 2 | ~15 | ~20 |
| Total reflections | 245867 | 66707 | 49321 | 48659 (367) |
| Unique reflections | 11791 (462) | 17206 (842) | 13983 (623) | 10998 (256) |
| Redundancy | 16.6 (11.4) | 3.9 (4.0) | 3.5 (1.7) | 4.4 (1.4) |
| Completeness (%) | 95 (96) | 98.5 (98.8) | 82.1* (36.4) | 64.0* (15.6) |
| Mean I/σ(I) | 6 (.61) | 6.41 (0.95) | 6.25 (1.56) | 8.83 (1.45) |
| R$_{merge}$ | 0.171 | 0.223 (1.388) | 0.186 | 0.185 |
| R$_{meas}$ | 0.177 | 0.270 (1.608) | 0.210 | 0.203 |
| R$_{pim}$ | 0.047 | 0.133 (0.795) | 0.095 | 0.080 |
| CC$_{1/2}$ | 0.99 (0.189) | 0.993 (0.396) | 0.984 (0.584) | 0.986 (0.683) |
| **Refinement** | | | | |
| Resolution range (Å) | 46.47–3.94 (4.12–3.94) | 30.0–3.0 (3.05–3.00) | 39.19–2.40 (2.49–2.40) | 39.19–2.36 (2.44–2.36) |
| R$_{work}$ | 0.271 (0.253) | 0.244 (0.328) | 0.237 (0.269) | 0.244 (0.307) |
| R$_{free}$ | 0.309 (0.343) | 0.270 (0.394) | 0.276 (0.358) | 0.299 (0.437) |
| Number of atoms | 5764 | 5979 | 3219 | 3115 |
| Protein | 5761 | 5812 | 3012 | 3012 |
| Ligand | 3 | 138 | 143 | 86 |
| Water | – | 29 | 64 | 17 |
| Protein residues | 786 | 774 | 398 | 398 |
| Ramachandran plot | | | | |
| Favored (%) | 754 (96.0) | 762 (98.4) | 389 (97.7) | 381 (95.7) |
| Allowed (%) | 32 (4.0) | 12 (1.6) | 9 (2.3) | 17 (4.3) |
| Outliers (%) | 0 (0.0) | 0 (0.0) | 0 (0.0) | 0 (0.0) |
| RMS(bonds) | 0.004 | 0.002 | 0.002 | 0.002 |
| RMS(angles) | 1.02 | 0.50 | 0.43 | 0.57 |
| Average B-factor | 90.3 | 61.27 | 73.6 | 50.1 |
| Protein | 90.3 | 61.29 | 73.1 | 49.9 |
| Ligand | 197.0 | 66.16 | 89.3 | 50.7 |
| Water | – | 34.08 | 63.7 | 78.8 |
| Number of TLS groups | 15 | 20 | 6 | 7 |

*The G223WdN34 dataset compiled from many small crystals is 99% complete to 3.4 Å, 95% complete to 3.0 Å, 90% complete to 2.85 Å, 80% complete to 2.7 Å, and 60% complete to 2.5 Å. The G223W ΔN34 apo dataset compiled from many small crystals is 96% complete to 4.5 Å, 92% complete to 3.4 Å, 77% complete to 3.0 Å, 66% complete to 2.85 Å, and 52% complete to 2.7 Å.

DOI: https://doi.org/10.7554/eLife.41124.005

G223W structure would clash with E176 on TM5. Consequently, the intracellular vestibule to the metal-binding site is highly constricted yet there is no aqueous pathway to the binding site from the external side (*Figure 2B*). Structural alignments with the inward-open *Staphylococcus capitis* (Sca) Nramp (*Ehrnstorfer et al., 2014*) and outward-open EcoNramp (*Ehrnstorfer et al., 2017*) also indicate an intermediate conformation for the G45R structure, albeit closer to the inward-open state (*Figure 2E* and *Figure 2—figure supplement 2D*), confirming our assignment as inward-occluded.

## Structure of G223W DraNramp provides first metal-bound outward-open state

The G223W structure (*Figure 2C*) represents an outward-open, metal-bound state that superimposes best with the outward-open EcoNramp structure (*Figure 2E* and *Figure 2—figure supplement 2E*). As predicted, the exogenous tryptophan lines a periplasmic-facing aqueous cavity leading to a bound $Mn^{2+}$ in the center of the transporter, with close helix packing below precluding metal passage to the cytoplasm. We also determined a G223W apo structure (*Table 1*), which lacks electron density attributable to metal substrate in the binding site (*Figure 2—figure supplement 2C*) but is otherwise similar to the metal-bound state ($C\alpha$ RMSD = 1.08 Å); hence we used the metal-bound structure for all further analyses. Compared to the inward-open and inward-occluded structures, in the outward-open state TM1b, TM6a, and the top of TM10 are splayed open, and loop 1–2 is displaced by ~4 Å, to form a wide aqueous pathway to the metal-binding site (*Figure 2H–I*). On the cytoplasmic side, TM4 and TM5 move significantly (by ~8 Å) straddling TM8 and approaching TM1a, while TM1a also approaches TM8 to fully shut the interior aqueous vestibule (*Figure 2H–I*).

## Comparisons of DraNramp structures define a scaffold and flexible regions

Based on overall superpositions of the three DraNramp structures, TMs 1, 4, 5, 6, and 10 show the largest displacements to switch metal-binding site accessibility (*Figure 2F–I*). The remaining TMs (2, 3, 7, 8, 9, and 11) would thus form a 'scaffold,' which adjusts to accommodate the more significant movements of the other five TMs (*Video 1*).

To more objectively compare the intramolecular rearrangements that occur during the transport cycle, we calculated difference distance matrices (*Richards and Kundrot, 1988*), averaged by TM, for each pair of structures (*Figure 3*). These matrices confirm that TMs 1, 4, 5, 6, and 10 undergo the most significant displacements relative to the rest of the protein between the different structures. But rather than moving as a rigid body such as proposed in the 'rocking bundle' model for LeuT (*Forrest and Rudnick, 2009*), these five TMs are also significantly displaced relative to each other.

## The G223W outward-open structure reveals a metal-coordination sphere distinct from the inward-open state's

Like other LeuT-family members, DraNramp relies on unwound regions of its TMs to bind substrates. A conserved DPGN sequence is non-helical in TM1—with the helix-breaking proline-glycine pair separating two metal-binding residues—while a conserved MPH sequence that includes the metal-binding methionine ends an unwound region in TM6 (*Figure 4A–C*). We used these two canonical motifs to generate an alignment of 6878 Nramp sequences (*Figure 2—source data 1*) and calculate the conservation of other important residues (*Figure 2—figure supplement 1C*). Interestingly, a third proline, P386 (83% conserved), enables the top of TM10 to swing to open the metal-binding site from the periplasm, while T228 (80%) on TM6 and N426 (99%) on TM11 stabilize the extended unwound TM6 region in the G223W structure (*Figure 4—figure supplement 1A*).

The inward-open ScaNramp structure revealed a metal-binding site consisting of three conserved sidechains corresponding to D56, N59, and M230 in DraNramp, and a backbone carbonyl of A227 (*Figure 4B*) (*Ehrnstorfer et al., 2014*). In outward-open DraNramp, a $Mn^{2+}$ binds both D56 and M230 (2.9 and 3.0 Å), with N59 slightly further away (3.4 Å) (*Figure 4A*). The increased unwinding of TM6a displaces the A227 carbonyl too far (6.5 Å) to coordinate the metal (*Figure 4—figure supplement 1A–B*). Instead, the A53 carbonyl coordinates the $Mn^{2+}$ (2.0 Å)—our structure is thus the first to implicate this residue in the metal transport cycle. Interestingly, A53 and A227 are at analogous positions within the two inverted repeats of the LeuT fold. Two waters (2.7 and 2.8 Å) complete a $Mn^{2+}$-coordination sphere. While the resolution remains too low to definitively define the

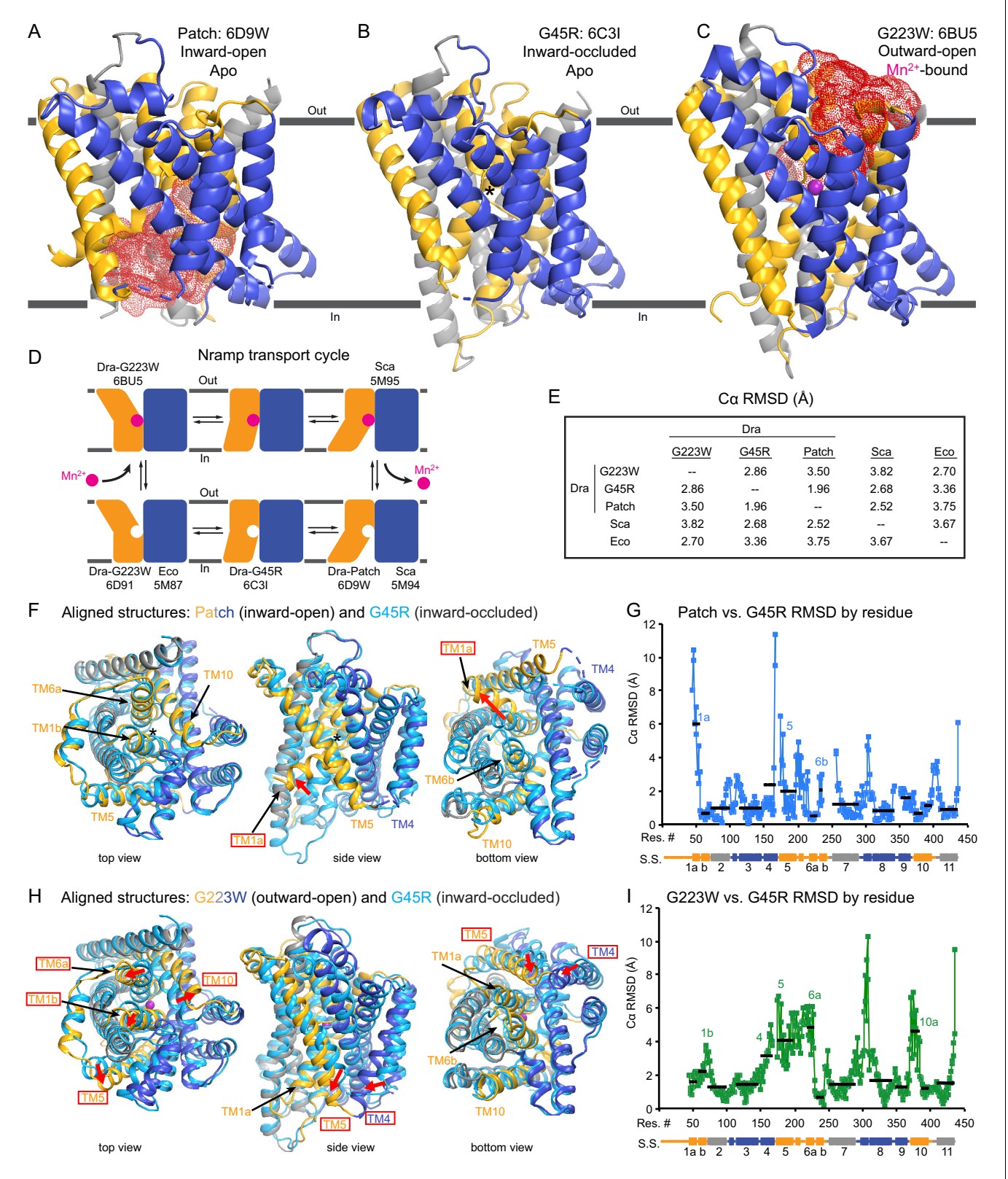

**Figure 2.** Crystal structures of DraNramp reveal two new conformations. (**A**) Updated DraNramp Patch mutant structure in an inward-open apo state with a wide intracellular aqueous vestibule (red mesh) (***Bozzi et al., 2016b***). TMs 1, 5, 6, and 10 are colored gold, TMs 3, 4, 8, and 9 blue, and TMs 2, 7, and 11 gray. (**B**) The G45R structure revealed an apo inward-occluded conformation with no substantial extracellular or intracellular aqueous pathway to

*Figure 2 continued*

the metal-binding site, denoted by an * in structures where no metal is present. (C) The G223W structure revealed an outward-open conformation with the physiological substrate $Mn^{2+}$ (magenta sphere) bound at the bottom of a substantial extracellular aqueous cavity (red mesh). All structures are viewed from within the membrane. (D) The complete Nramp transport cycle likely consists of at least six distinct conformational states. Including the structures of *S. capitis* (Sca) (*Ehrnstorfer et al., 2014*) and *E. coleocola* (Eco) (*Ehrnstorfer et al., 2017*) homologs, we now have structures of five of these conformations. (E) Pairwise RMSD values for the superposition of the 355 Cα atoms present in all three DraNramp structures, and ScaNramp and EcoNramp. G223W superimposes best with the outward-open EcoNramp, validating our mutagenesis strategy to obtain an outward-open conformation. G45R superimposes better with the inward-open Patch mutant than outward-open G223W, suggesting it represents an inward-occluded conformation. (F) Superposition of Patch mutant (gold, gray, and blue) and G45R (cyan) and (G) RMSD calculated at each Cα. These structures superimpose well except for TM1a. TM1a's significant displacement from the inward-occluded state (G45R) to the inward-open state (Patch) is highlighted with a red arrow. (H) Superposition of G223W (gold, gray, and blue) and G45R (cyan) and (I) RMSD calculated at each Cα. The most significant rearrangements involve TMs 1b, 4, 5, 6a, and 10. These significant displacements from the inward-occluded state (G45R) to the outward-open state (G223W) are highlighted with red arrows. In panels G and I, black lines indicate average RMSD for each TM, with TMs 1, 6, and 10 divided into two halves. In panels F and H, the central view is rotated 45° along the vertical axis from the view in panels A-C, while the left and right views are 90° rotations of that central view. See also *Figure 2—figure supplements 1* and *2*.

DOI: https://doi.org/10.7554/eLife.41124.006

The following source data and figure supplements are available for figure 2:

**Source data 1.** Nramp sequence alignment.
DOI: https://doi.org/10.7554/eLife.41124.009
**Figure supplement 1.** Secondary structure and primary sequence of DraNramp.
DOI: https://doi.org/10.7554/eLife.41124.007
**Figure supplement 2.** Electron density maps for new crystal structures and structural comparisons to other Nramp homologs.
DOI: https://doi.org/10.7554/eLife.41124.008

coordination geometry, the electron density is consistent with $Mn^{2+}$ interacting with both D56 oxygens and thus seven total ligands—rare but not unprecedented for $Mn^{2+}$ (*Barber-Zucker et al., 2017*; *Glasfeld et al., 2003*). An ordered water network expands into the external vestibule as part of the extended metal coordination sphere (*Figure 4—figure supplement 1A*). A water is also tethered to the conserved H232 directly below the metal-binding M230, perhaps poised to hydrate the cation upon conformational change.

## The G45R inward-occluded structure suggests potential metal-binding role for conserved Q378

The inward-occluded G45R binding site contains no metal. The A53 carbonyl is farther from the other metal-binding residues than the A227 carbonyl (*Figure 4C*). This is consistent with a model in which Nramp metal transport involves a switch of ligands, perhaps with the A53 and A227 carbonyls both coordinating the metal substrate in an as-yet-uncaptured intermediate state.

In G45R, flexing of TM10 above its P386 pivot shifts Q378 (86% conserved, with another 11% of sequences, including HsNramp2, having an N at this position) ~5 Å to within hydrogen-bonding distance of metal-binding A227 and D56, perhaps stabilizing the negative charge on a deprotonated D56 during the empty transporter's return to outward-open. While Q378 does not

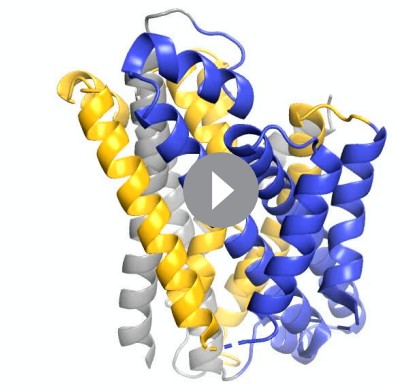

**Video 1.** Internal rearrangements during DraNramp conformational changes. Morph of the structure of DraNramp based on a global superimposition of the three captured conformational states. TMs 1, 5, 6, and 10 are colored gold, TMs 3, 4, 8, and 9 blue, and TMs 2, 7, and 11 gray, viewed in the membrane plane and the intracellular face pointing down. The morph starts from the G223W outward-facing conformation, transitions to the G45R occluded state, then to the Patch mutant inward-open state. The view then shifts to the intracellular face, followed by the external face, continuing to alternate back and forth through these three conformations.

DOI: https://doi.org/10.7554/eLife.41124.010

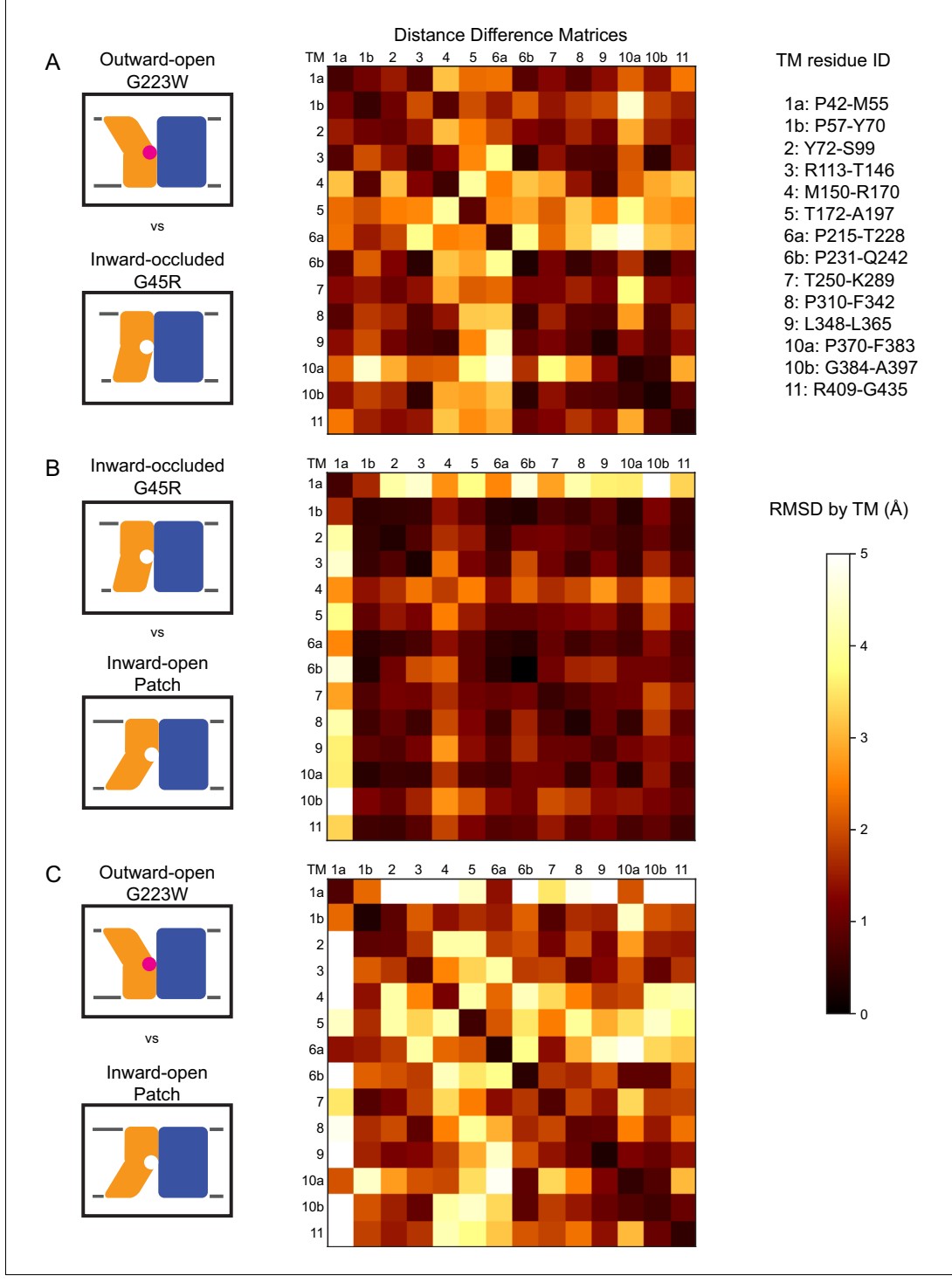

**Figure 3.** Distance difference matrices illustrate internal rearrangements during DraNramp conformational change. We calculated the pairwise Cα – Cα distances to obtain a 398 × 398 distance matrix for each of the three DraNramp structures. To compare two structures, we generated distance difference matrices (*Richards and Kundrot, 1988*), subtracting one distance matrix from the other. To simplify the data and focus on relative movements between helices, we then calculated the root-mean-squared deviations of submatrices, grouping residues within each helical segment. We used a total of 14 segments (top right), breaking TMs 1, 6, and 10 into two separate segments before and after their helix-breaking elements. The resulting 14 × 14 matrices compare (**A**) outward-open G223W and inward-occluded G45R, (**B**) inward-occluded G45R and inward-open Patch mutant, and (**C**) outward-open G223W and inward-open Patch mutant. Pairs of helices that remain stationary relative to each

*Figure 3 continued on next page*

*Figure 3 continued*

other have RMSD values close to 0 indicated by darker colors. In contrast, pairs of helices that rearrange significantly relative to each other have larger magnitude values indicated by lighter coloring in the heat map. These matrices show that (**A**) TMs 4, 5, 6a, and 10a undergo the greatest displacement relative both to the rest of the protein and to each other in the conformational change from inward-occluded to outward-open. (**B**) In contrast, the conformational change from inward-open to inward-occluded consists primarily of the large displacement of TM1a, with the rest of the protein remaining mainly stationary. (**C**) The comparison of the outward-open and inward-open states is essentially a sum of the two previous comparisons, with the large TM1a displacement added to the significant movements of TMs 4, 5, 6a, and 10a. The Python code used to perform the calculations and generate the resulting plots is available at GitHub: https://github.com/GaudetLab/coarse-grained-DDMP.

DOI: https://doi.org/10.7554/eLife.41124.011

bind the metal substrate in either outward DraNramp (7.9 Å) or inward ScaNramp (*Ehrnstorfer et al., 2014*) (4.5 Å), its position in the G45R occluded intermediate suggests it may transiently bind during the transport process. Indeed, two independent molecular dynamics (MD) simulations of the inward-open ScaNramp showed a metal interaction with the Q378 oxygen (*Bozzi et al., 2016a*; *Pujol-Giménez et al., 2017*), and mutations at this position impaired metal transport in HsNramp2 (*Pujol-Giménez et al., 2017*).

To test the importance of the three conserved sidechains that coordinate $Mn^{2+}$ (D56, N59, and M230; *Figure 4A–B*) and Q378 to metal transport, we purified a panel of mutants and reconstituted them into proteoliposomes (*Figure 4D*). D56A and D56N eliminated $Mn^{2+}$ and $Cd^{2+}$ transport (*Figure 4E–F* and *Figure 4—figure supplement 1C–D*), confirming the importance of D56. N59A severely and N59D moderately reduced transport of both metals (*Figure 4E–F* and *Figure 4—figure supplement 1C–D*). Both M230A and M230T transport both metal substrates (*Figure 4—figure supplement 1C–D*), but with lower apparent affinity than WT (*Figure 4E–F*). Consistent with our previous findings, removing M230—the lone sulfur-containing metal-binding residue—affects $Cd^{2+}$ more than $Mn^{2+}$ transport, reflecting the importance of the semi-covalent interaction $Cd^{2+}$ can form with sulfur (*Bozzi et al., 2016a*). Lastly, Q378S and Q378N both preserved significant transport of both metals (*Figure 4—figure supplement 1C–D*). However, these mutants increased the $K_M$ for $Mn^{2+}$—but interestingly not $Cd^{2+}$ (*Figure 4E–F*). That the native glutamine is essential for efficient transport of the biological substrate $Mn^{2+}$ but dispensable for $Cd^{2+}$ uptake (*Figure 4—figure supplement 1E*) suggests that the two metals interact differently with their surrounding ligands during the transport process, corroborating the differential effects of M230 mutations.

## DraNramp proton transport does not require large conformational change

We reconstituted G45R and G223W into proteoliposomes to assess their metal and proton transport (*Figure 5A*). Consistent with our in vivo findings, neither G45R nor G223W transported $Mn^{2+}$, $Cd^{2+}$, or $Co^{2+}$ (*Figure 5B* and *Figure 5—figure supplement 1A–B*). Surprisingly, in the presence of a favorable negative membrane potential established using $K^+$ gradients and the $K^+$-ionophore valinomycin, G223W enabled larger basal $H^+$ influx than WT, while G45R had no $H^+$ flux (*Figure 5C*). Interestingly, while $Mn^{2+}$ transport stimulated $H^+$ uptake for WT, adding $Mn^{2+}$ did not further stimulate either G45R or G223W (*Figure 5C*). These results suggest that Nramp metal and proton transport can proceed via separate routes, with proton transport requiring only that the protein sample the outward-open state.

To test this new hypothesis, we reconstituted the A53C and A61C mutants, with single cysteines located just below or above the metal-binding site respectively (*Figure 5—figure supplement 1D*). Both retain significant metal transport (*Figure 5E,G* and *Figure 5—figure supplement 1E–F*), and can be targeted with cysteine-specific modifiers to post-translationally add bulky and/or charged wedges to impede conformational change (*Figure 5D* and *Figure 5—figure supplement 1C*). Charged, and thus membrane-impermeable, MTSET nearly eliminated metal transport by A61C, while uncharged NEM or MTSEA moderately impaired or did not affect transport, respectively (*Figure 5E* and *Figure 5—figure supplement 1E*), a result consistent with our previous in vivo

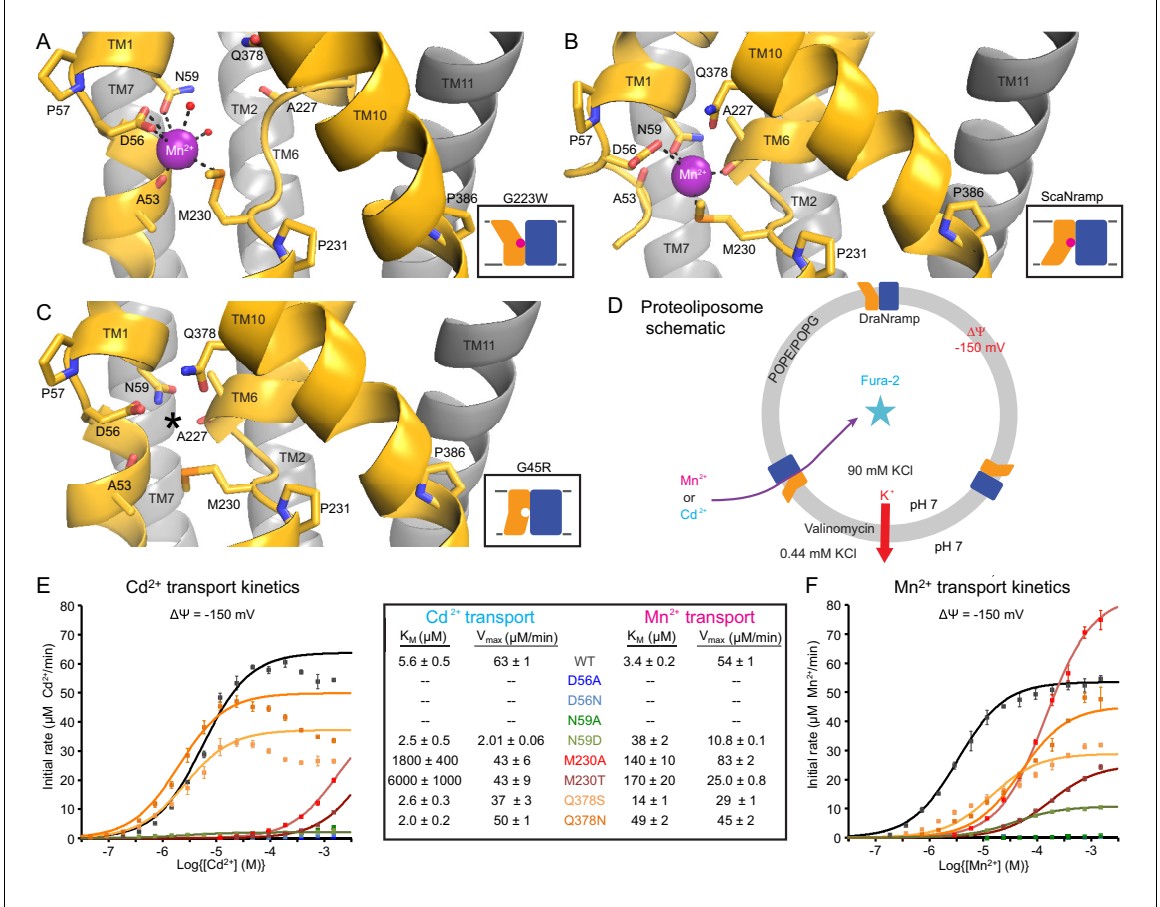

**Figure 4.** Metal-coordinating interactions vary across conformational states. (**A**) In the G223W outward-open structure D56, N59, M230 and the A53 carbonyl, along with two water molecules, coordinate $Mn^{2+}$. (**B**) In the inward-open ScaNramp structure (PDB: 5M95, DraNramp residue numbering) D56, N59, M230, and the A227 carbonyl coordinate $Mn^{2+}$. (**C**) In the G45R inward-occluded structure D56, N59, M230, Q378, and the A53 and A227 carbonyls are all close, suggesting that all six could simultaneously coordinate the metal substrate in a hypothetical similar $Mn^{2+}$-bound conformation. Three conserved helix-breaking prolines on TM1 (P57), TM6 (P231), and TM10 (P386) confer the flexibility needed for the metal-binding site. The TM6 unwound region is extended in the outward-open state, while TM10 bends more dramatically at P386 in the inward-oriented states to close outside access to the metal-binding site. For clarity, TMs 3, 4, 5, 8, and 9 are omitted in panels A-C. (**D**) Schematic for liposome transport assay to measure $Mn^{2+}$ or $Cd^{2+}$ uptake. (**E–F**) Plots of initial rates vs. (**E**) $Cd^{2+}$ or (**F**) $Mn^{2+}$ concentration used to determine Michaelis-Menten constants (Table inset) for WT DraNramp and binding-site mutants. All mutants significantly impaired $Mn^{2+}$ transport at lower $[Mn^{2+}]$, while only Q378 mutants did not severely impair $Cd^{2+}$ transport. Data are averages ± S.E.M. (n = 3); errors on $K_M$ and $V_{max}$ reflect uncertainty of the fit (shown as solid lines on the graphs) to the data. For WT and Q378 mutants we observed a reduced transport rate at high $[Cd^{2+}]$, which we did not include in the $K_M/V_{max}$ fitting. These data are intriguing and remain unexplained, as we do not observe a similar effect for $Mn^{2+}$, and we do not see an obvious second regulatory metal-binding site in our current structures of DraNramp. See also **Figure 4—figure supplement 1**.

DOI: https://doi.org/10.7554/eLife.41124.012

The following figure supplement is available for figure 4:

**Figure supplement 1.** Extended TM6 unwound region enables metal-binding in outward-open DraNramp.

DOI: https://doi.org/10.7554/eLife.41124.013

findings that adding steric bulk, but not formal charge, is tolerated at this position (*Bozzi et al., 2016b*). In addition, MTSET-treated A61C replicated G223W's $H^+$-transport behavior, with higher $H^+$ uniport compared to unmodified transporter, but little stimulation by $Mn^{2+}$ (*Figure 5F*). In contrast MTSET had no effect on $Mn^{2+}$ and $Cd^{2+}$ transport by A53C, but membrane-permeable MTSEA and NEM both eliminated transport (*Figure 5G* and *Figure 5—figure supplement 1F*). Regarding $H^+$ uptake, MTSEA- and NEM-treated A53C resembled G45R, with no basal uniport or $Mn^{2+}$ stimulation, while again unmodified and MTSET-treated A53C both behaved similarly to WT (*Figure 5H*). These findings show that essentially all activity in proteoliposomes comes from outside-out

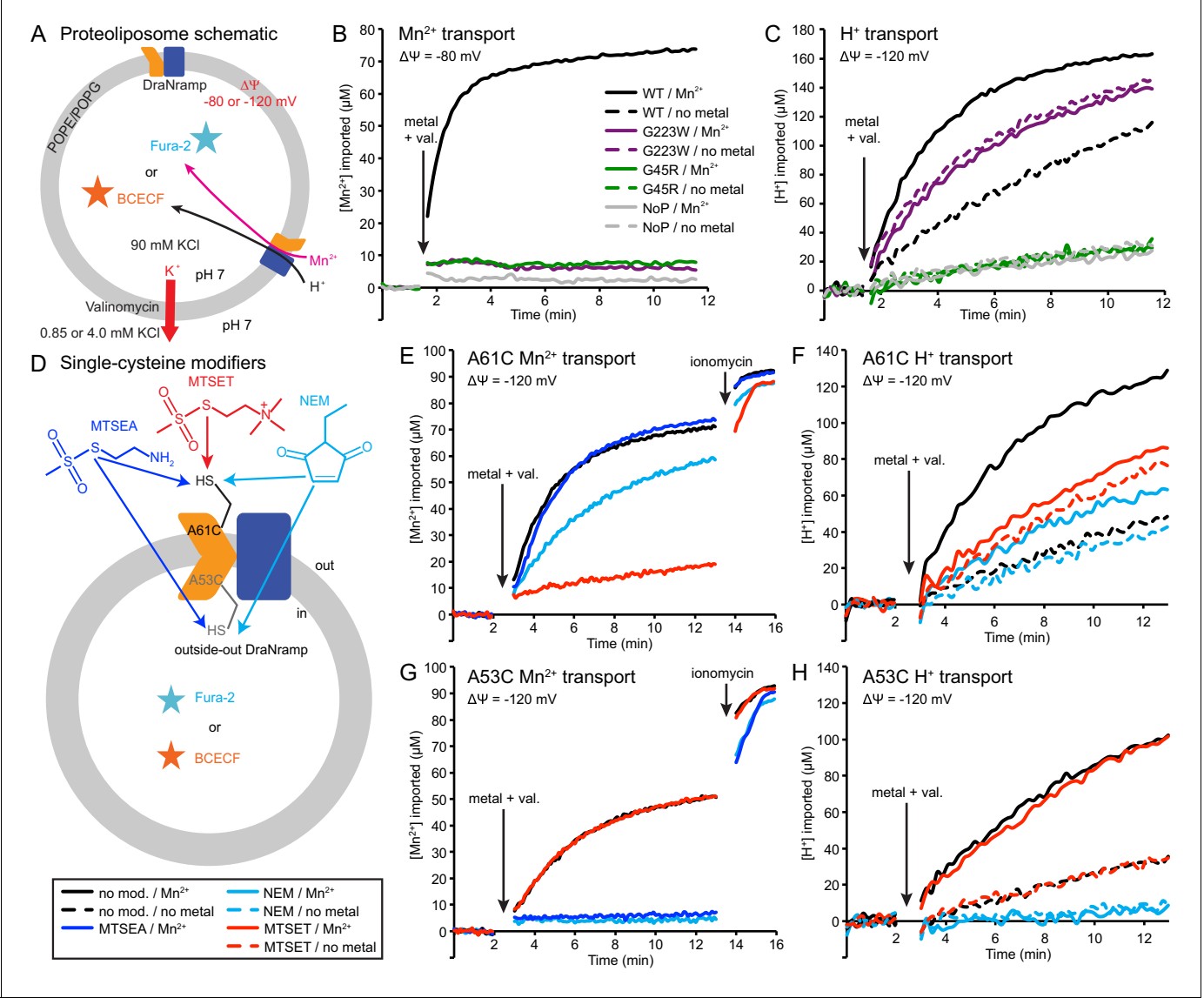

**Figure 5.** Outward-locked mutants retain H⁺ transport ability. (**A**) Proteoliposome assay schematic for monitoring M²⁺ and H⁺ import. Membrane potential (ΔΨ) was established using K⁺ gradients and valinomycin. (**B**) Mn²⁺ uptake over time; WT enabled robust uptake while G45R and G223W showed no activity. NoP = No Protein control liposomes. (**C**) H⁺ uptake over time; WT enabled significant H⁺ uptake without metal (H⁺ uniport), further stimulated by Mn²⁺. G223W showed a larger H⁺ uniport rate, but Mn²⁺ provided no enhancement; G45R showed no H⁺ transport activity. Traces are representative of four experiments. (**D**) Schematic for in vitro cysteine modification of A53C (inside accessible) and A61C (outside accessible). MTSEA and NEM are membrane permeable, whereas permanently-charged MTSET is not. (**E,G**) Mn²⁺ and (**F,H**) H⁺ transport over time in the presence or absence of cysteine modifying agents. Adding ionomycin allows divalent cation entry to achieve maximum signal. Traces are representative of three experiments. See also *Figure 5—figure supplements 1*, *2* and *3*.

DOI: https://doi.org/10.7554/eLife.41124.014

The following figure supplements are available for figure 5:

**Figure supplement 1.** Conformational locking impairs metal transport.
DOI: https://doi.org/10.7554/eLife.41124.015

**Figure supplement 2.** DraNramp inserts with a random orientation in proteoliposomes.
DOI: https://doi.org/10.7554/eLife.41124.016

**Figure supplement 3.** Divalent metals have little-to-no effect on BCECF fluorescence.
DOI: https://doi.org/10.7554/eLife.41124.017

DraNramp. MTSET, which should inhibit any inside-out A53C, did not affect transport (*Figure 5G* and *Figure 5—figure supplement 1F*), indicating that inside-out transporters contribute negligibly to the total activity in this assay. Consistently, while MTSET treatment spares inside-out A61C from labeling, it nevertheless nearly eliminated metal transport (*Figure 5E* and *Figure 5—figure supplement 1E*), further supporting the assertion that outside-out WT-like transporters provided most of the detected transport activity. To confirm that a mix of inside-out and outside-out transporters were indeed present in proteoliposomes, we assessed the susceptibility of DraNramp to thrombin cleavage at a naturally-occurring site (*Gallwitz et al., 2012*) in the protein's non-conserved, disordered N-terminal region (*Figure 5—figure supplement 2A–C*). While thrombin fully cleaved DraNramp in detergent, in proteoliposomes the cleaved protein population plateaued at ~50% (*Figure 5—figure supplement 2D–F*). This cleaved portion likely corresponds to inside-out oriented protein with an exposed N-terminal region, with the remaining ~50% of protein therefore outside-out oriented with the N-terminus inside the liposome and thus protected from thrombin cleavage (*Tsai et al., 2012*; *Tsai et al., 2013*).

In summary, these experiments with permanently-locked crystallization constructs or chemically-locked cysteine mutants demonstrated that while metal transport requires complete conformational cycling, proton transport does not require large-scale conformational change and can proceed through DraNramp's outward-open state but not its inward-open state. In addition, metal transport through DraNramp is much more efficient in the outside-to-inside direction than in the inside-to-outside direction under the physiological-like conditions set up in our in vitro assay.

## Conserved salt-bridge network provides potential proton pathway to cytoplasm

Our in vitro results suggested that proton transport occurs via a pathway separate from the intracellular metal-release vestibule, which remains closed to bulk solvent in the proton-transporting G223W mutant. Below the metal-binding site begins a network of highly-conserved hydrophilic residues, including at least seven potentially protonatable sidechains, that leads from the metal-binding D56 through a tight corridor between TMs 3, 4, 8, and 9 to the cytoplasm (*Figure 6A–C*). In contrast to the external and intracellular vestibules proposed as metal entrance and release pathways, the helices and residues within this polar network undergo little rearrangement between the three DraNramp structures, except the intracellular end of TM4 (*Figures 2F–I* and *3*).

Highly-conserved residues surrounding the metal-binding D56 include H232 on TM6b (100% as defined) and E134 (TM3, 98%)—which have each been proposed as the Nramp proton transfer point (*Ehrnstorfer et al., 2017*; *Pujol-Giménez et al., 2017*). Across from E134 lies a conserved salt-bridge pair: D131 (TM3, 93%) and R353 (TM9, 78%). Approximately 9 Å below, a second conserved salt-bridge, E124-R352 (94% and 87%), links the same two helices. This network could provide the route for proton uniport in the outward-open conformation.

## Two conserved aspartates anchor the DraNramp proton transport pathway

To assess whether these residues could be proton carriers, we calculated predicted $pK_a$ values for our outward-open and inward-occluded structures (*Figure 6D*) (*Dolinsky et al., 2004*). Surprisingly, D56 is the only residue with a $pK_a$ in the ideal 6–7 range to facilitate proton exchange at a typical external pH. About 4 Å from D56, E134's high $pK_a$ (~11.5) indicates a near permanently-protonated state, while H232—4 Å below E134 and 7 Å from D56—has too low a $pK_a$ (~3.5) to easily protonate, as does H237 (~4.0) further down TM6b. While E134 and H232 have separately been suggested as the Nramp proton-binding site (*Ehrnstorfer et al., 2017*; *Pujol-Giménez et al., 2017*), our $pK_a$ predictions suggest otherwise, as maintaining a formal change, especially on a histidine, would not be favorable in the protein core. In addition, previous studies showed the analogous E-to-Q mutant in EcoNramp maintained WT-like proton transport ability (*Ehrnstorfer et al., 2017*) as did the analogous H-to-A mutant in rat Nramp2 (*Mackenzie et al., 2006*), which argues against those two residues as essential transfer points if the Nramp family shares a common transport mechanism. Within the TM3-TM9 salt-bridge network, R352 and R353 are likely protonated and positively charged, while their respective partners E124 and D131 are likely deprotonated and negatively charged. The predicted $pK_a$ values of D131 and E124 indicate their amenability to protonation. Indeed, as D56's

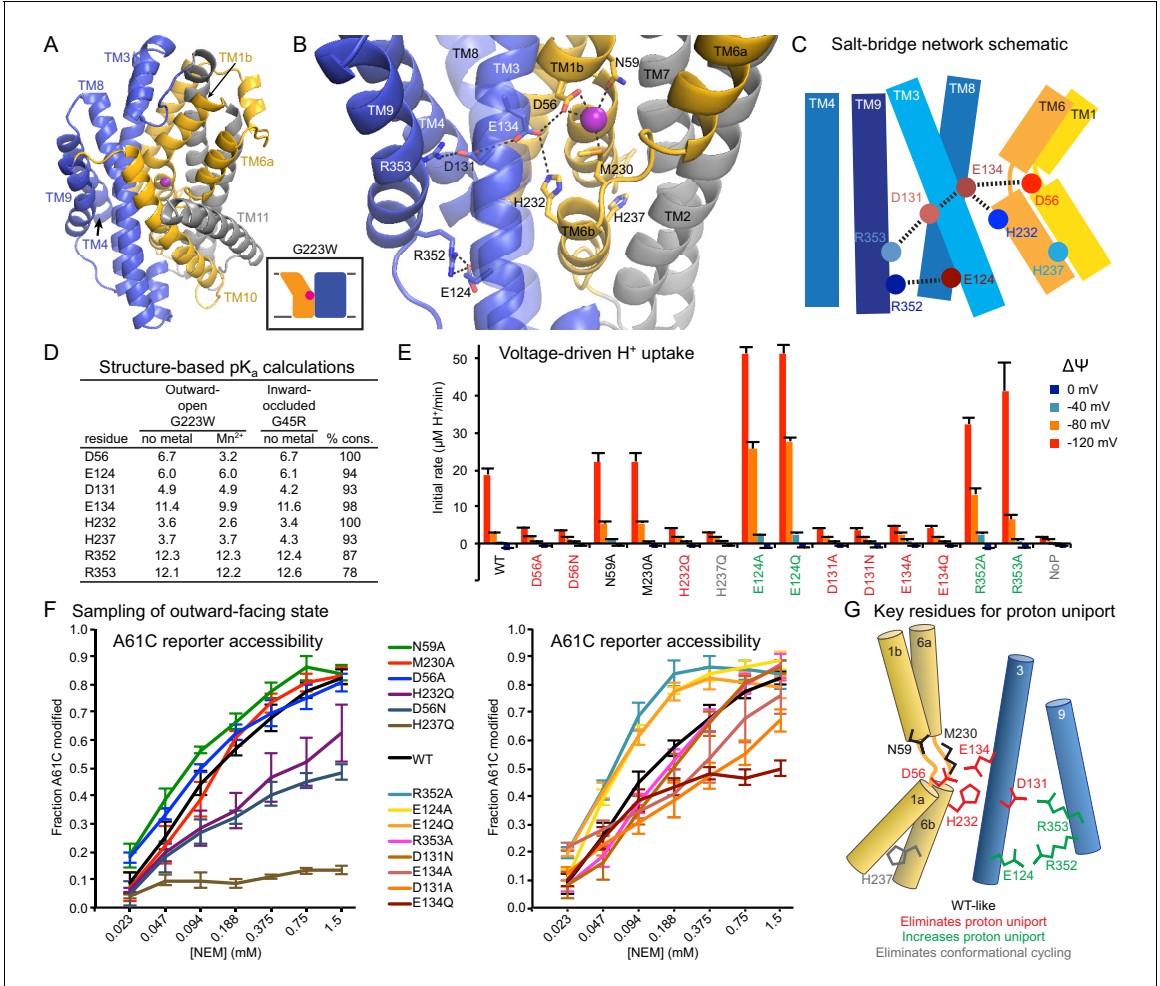

**Figure 6.** Conserved salt-bridge network provides pathway for proton uniport. (**A**) View from an angle above the membrane looking down into the extracellular aqueous cavity in the outward-open G223W structure. (**B**) Zoomed-in view (with TM10 and TM11 omitted for clarity) and (**C**) schematic showing a network of conserved protonatable residues that originates from the metal-binding site and extends into the cleft between TMs 3, 4, 8, and 9. H232 and E134 abut metal-binding M230 and D56, providing a connection to the D131-R353 and R352-E124 salt-bridge pairs. (**D**) pKa estimates by PROPKA (**Dolinsky et al., 2004**) from high-resolution DraNramp structures. (**E**) Initial rates of basal H⁺ uptake at various ΔΨs (averages ± S.E.M.; n ≥ 4). Mutations to D56, E134, H232, and D131 eliminated H⁺ uniport observed for WT. N59 and M230 mutants retained significant H⁺ uptake. Mutations to E124, R352, and R353 enhanced H⁺ uniport. (**F**) Dose-response curve of outward-open reporter A61C modification versus NEM concentration. All mutants except H237Q sampled the outward-open state, which is required for H⁺ transport to occur (**Figure 5**). Data are averages ± S.E.M. (n ≥ 3). (**G**) Salt-bridge network schematic shows clustering of four residues required for H⁺ uniport (red) as well as three residues that may restrain this flux (green). See also **Figure 6—figure supplement 1**.

DOI: https://doi.org/10.7554/eLife.41124.018

The following figure supplement is available for figure 6:

**Figure supplement 1.** Sample traces illustrate mutant perturbations of voltage-driven proton uniport.

DOI: https://doi.org/10.7554/eLife.41124.019

predicted pKa drops to 3.2 with Mn²⁺ bound, D131 becomes the best candidate to receive a proton.

We observed three distinct voltage-driven H⁺ transport phenotypes within a panel of mutants to highly-conserved residues (**Figure 6E** and **Figure 6—figure supplement 1**). First, removing either metal-binding residue N59 or M230 had little effect. Second, neutralizing any member of the D56-E134-H232-D131 network or the H237Q mutation drastically reduced H⁺ transport. Third, mutating any of E124, R352, R353—farthest from D56—increased H⁺ uniport across multiple voltages.

Outward-reporter A61C accessibility (**Figure 1C**) is consistent with each mutant sampling the outward-open state needed for proton transport (**Figure 5**), ruling out a conformation-locking

explanation for the loss-of-function mutants. While some mutations perturbed the transporter's conformational preference, A61C remained at least somewhat accessible in all cases except H237Q (*Figure 6F*).

In summary, based on our $H^+$ transport measurements, structure-calculated residue $pK_a$ values, and prior studies using mutants of other Nramp homologs, D56 is the likely initial protonation point, with E134 and H232 positioned to chaperone the proton transfer to D131, while R352, R353, and E124 restrain this process (*Figure 6G*).

## Discussion

We propose a structure-based model for conformation cycling in DraNramp (*Figure 7A–B*). Starting from the outward-open state seen in our G223W structure (*Figure 7A*, left panel), metal binding (and perhaps resulting proton entry into its release pathway) may trigger bulk conformational rearrangement (see below for details). To close the external vestibule, TM6a, TM10, and to a lesser extent TM1b move closer to each other above their respective non-helical hinge regions, with the TM6a movement propagated through the TM5-6 linker to reorient TM5 and thus begin to open the inner gate. From this transient occluded conformation similar to our G45R structure (*Figure 7A*, middle panel), additional movement of TM4-TM5 allows TM1a to bend upward to fully open the inner gate, enabling solvent access to and release of the metal, as the protein achieves a state similar to the Patch mutant DraNramp structure (*Figure 7A*, right panel) (*Bozzi et al., 2016b*). Analogously, to return to the outward-open state and complete the transport cycle, TM1a swings in to reach a conformation similar to G45R, then TM4-TM5 fully close on TM1a to seal the cytoplasmic vestibule while TM1b, TM6a, and TM10 separate to open the external vestibule.

Our in vitro assays showed that while DraNramp metal transport requires sampling of both outward- and inward-open states, proton uniport occurs in sterically outward-locked constructs (*Figures 1* and *5*). This supports a model where protons and metal travel through distinct pathways on the cytoplasmic side of the protein (*Figure 7C–D*), such that proton uniport is a feature of DraNramp's outward-open state, whereas metal transport requires bulk rearrangement. In contrast, both protons and metal likely enter through the same aqueous pathway, as inward-locked proteins do not transport either substrate. From structure-based $pK_a$ calculations (*Figure 6D*), $H^+$-transport data for a large panel of mutants to conserved protonatable residues in DraNramp (*Figure 6E*), and previous mutagenesis studies with other Nramp homologs (*Ehrnstorfer et al., 2017*; *Mackenzie et al., 2006*; *Pujol-Giménez et al., 2017*), we propose that proton uniport occurs via a network of conserved protonatable residues leading from D56 in the metal-binding site to D131 in a salt-bridge network between TMs 3, 4, 8, and 9. This proton pathway is accessible in the outward-open state, thus enabling the well-documented proton uniport (*Chen et al., 1999*; *Gunshin et al., 1997*; *Mackenzie et al., 2006*; *Nelson et al., 2002*; *Xu et al., 2004*). The proton uniport—common to the general Nramp family—that occurs under physiological conditions wastes electrochemical energy by dissipating the transmembrane proton gradient without contributing to metal uptake. Considering the relatively low abundance and slow kinetics of Nramp transporters, this proton uniport property may be an evolutionarily-tolerated consequence of the transporter's design, or it could instead confer an as-yet-undetermined functional advantage. The predicted protonation of D56 and subsequent transfer to D131, mediated through an E134/H232-stabilized transition state, may however serve to restrain $H^+$ entry. For metal-stimulated proton transport, $Mn^{2+}$ binding likely stimulates proton transfer into the same salt-bridge network, perhaps by directly ejecting a proton from D56 in the metal-binding site. Indeed, in a separate study we show that neutralizing mutations to the same four residues that eliminated $H^+$ uniport (*Figure 6E*) also eliminated (D56, D131, H232) or severely reduced (E134) $H^+$ fluxes in the presence of added $Mn^{2+}$, with mutants to D131, E134, and H232 retaining significant metal transport despite a lack of proton transport (*Bozzi et al., 2018*). However, the precise order of events for proton and metal transport, including whether it is indeed a thermodynamically coupled symport mechanism, remains undetermined, and additional transport mechanisms are possible.

In our G223W structure, two water molecules coordinate $Mn^{2+}$: one lies between the metal and A227's carbonyl, the other the metal and Q378 (*Figures 4A* and *7E*). We propose that after $Mn^{2+}$ binds to D56, M230, A53, and N59 as in our G223W structure, the A227 carbonyl and Q378 both displace the two waters as the outer gate closes. DraNramp would thus reach a fully dehydrated

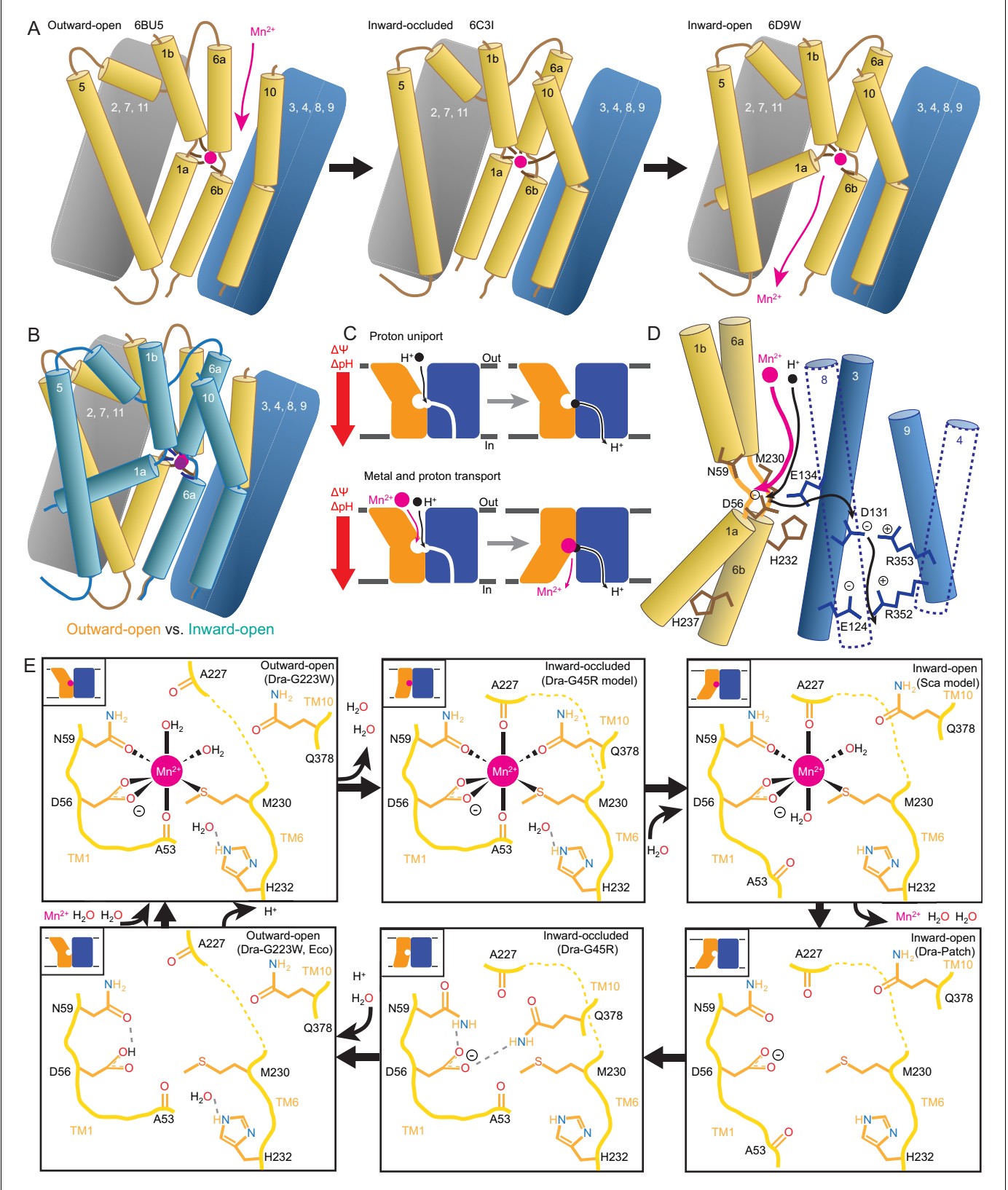

**Figure 7.** Mechanistic models for conformational change, proton transport, and metal transport. (**A**) Model of conformational change: The metal ion reaches the outward-open binding site via the vestibule between TMs 1b, 6a, 10, 3, and 8. TM6a and 10 bend inward to close the vestibule in the

*Figure 7 continued*

inward-occluded conformation. TM6a's inward bend pulls on TM5 via extracellular loop 5–6 to initiate opening of the inner gate. Finally, TM1a swings away from TM6b to open the intracellular vestibule into which the metal diffuses. (**B**) Superposition of outward- (gold) and inward-open (teal) conformations illustrate key movements by TMs 1, 5, 6, and 10. (**C**) Overall DraNramp transport model under prevailing physiological conditions (negative inside membrane potential, acidic-outside pH gradient): While metal transport requires complete conformational cycling, proton uniport occurs through the outward-open state. Metal-stimulated proton transport may follow the same pathway as proton uniport. (**D**) Proton transport model: A proton transits the external vestibule to reach the binding site near the membrane center and initially binds to D56. Metal enters through the same passageway, ejecting the proton, which then passes to D131, with H232 and E134 facilitating the transfer. The proton ultimately reaches the cytoplasm through the polar network between TMs 3, 4, 8, and 9, while the metal must await a bulk conformational change that opens a separate pathway between TMs 1a, 2, 5, 6b, 7, and 8. Proton uniport follows the same route, with the D56-D131 transfer occurring at a slower rate without metal substrate. (**E**) Model for metal coordination during the transport process: the metal initially binds in the outward-open state to D56, N59, M230 sidechains and the A53 carbonyl, shedding all but two water ligands. As the outward metal-permeation pathway closes, Q378 and the A227 carbonyl displace the waters so that the metal is fully coordinated by six amino acids in the inward-occluded state. As the inner gate opens, Q378 and the A53 carbonyl withdraw and are replaced by waters. The metal is then released into the cytoplasmic aqueous vestibule. To facilitate the return to the outward-open state, Q378 and N59 donate hydrogen bonds to negatively-charged D56 in the inward-occluded return conformation seen in our G45R structure. Finally, as the transporter returns to an outward-open state, it may bind a proton in preparation for metal-binding. For the inward-open metal-bound model for DraNramp, we altered the reported coordination from the $Mn^{2+}$-bound ScaNramp relatively low-resolution (3.4 Å) crystal structure (PDB: 5M95) by adding two metal-coordinating waters, as well as switching the aspartate coordination to bidentate and the asparagine coordination to monodentate to be consistent with our G223W structure.

DOI: https://doi.org/10.7554/eLife.41124.020

metal-bound state not yet visualized but which may resemble our apo G45R inward-occluded structure (*Figure 7E*). Next, as the inner gate opens, the A53 carbonyl would exchange with a nearby water—such as the one bound to H232 in our G223W structure—as would Q378, to yield an inward-open metal-bound state similar to the ScaNramp structure (*Figures 4B* and *7E*) (*Ehrnstorfer et al., 2014*). In this conformation the $Mn^{2+}$-coordination sphere would include four residues and two waters—analogous to the G223W structure—thus facilitating eventual metal release. The proposed transition from four to six to four $Mn^{2+}$-coordinating residues could help preferentially stabilize the occluded transition state (*Shilton, 2015*) through the free energy (entropy-driven) gains of releasing the two water ligands. Furthermore, the rearrangements needed to achieve the hypothetical intermediate six-residue $Mn^{2+}$ coordination—the helical extension and inward movement of TM6a, and the toppling of TM10's top half—also close the external vestibule, providing a potential mechanistic link between local metal-coordination changes and bulk conformational change. To return to the outward-open state, the transporter must pass through an apo-occluded state as seen in the G45R structure, in which the N59 and Q378 sidechains reorient to stabilize D56 in the absence of the divalent cation carried during the outward-to-inward transition (*Figures 4C* and *7E*). As the transporter reaches the outward-open state seen in EcoNramp (*Ehrnstorfer et al., 2017*) and our apo G223W structure, a protonation event at D56 may prime the binding site to receive another incoming metal ion. Future molecular dynamic simulations and/or experiments will be essential to test these predictions.

The mechanism described above for the DraNramp transport cycle—developed from structures of the same Nramp homolog in three distinct conformations and supported by metal and proton transport data—differs significantly from those previously observed for other LeuT-fold transporters. Mhp1, BetP, and (to a lesser degree) LeuT generally obey a 'rocking bundle' model in which the rigid-body movement of four TMs that contain the primary substrate binding site (1, 2, 6, and 7) against the remaining TMs (3-5, 8-10) leads to conformational change (*Forrest and Rudnick, 2009*; *Kazmier et al., 2017*; *Shi, 2013*). In DraNramp TMs 4, 5, and 10 join TMs 1 and 6 to form the substrate-binding 'mobile domain,' while TMs 2 and 7 join the remaining TMs as part of the scaffold. Furthermore, the mobile helices do not move as rigid bodies, as conserved helix-breaking motifs free TMs 1a, 6a, and the top of TM10 to move independent of TMs 1b, 6b and the bottom of TM10. In contrast, the fully-helical TM5 wholly reorients, and may thus coordinate the opening and closing of the inner and outer gates, connecting TMs 1a, 4, and 6b with TM6a (*Figures 2* and *7A–B*).

In comparison to other APC superfamily members, the large TM1a displacement in DraNramp most closely resembles its dramatic movement in LeuT (*Krishnamurthy and Gouaux, 2012*). Gating roles for TMs 5 and 10 have been ascribed for BetP, Mhp1, and MhsT (*Malinauskaite et al., 2014*;

*Ressl et al., 2009*; *Shimamura et al., 2010*), although not as extensive as we propose here in DraNramp. Not surprisingly, the DraNramp conformational changes are most similar to those predicted by comparing structures of two other bacterial Nramp homologs in complementary conformations (*Ehrnstorfer et al., 2014*; *Ehrnstorfer et al., 2017*), suggesting conservation within the Nramp clade of the LeuT-fold family.

Whereas the distinct conformational changes of DraNramp demonstrate the diverse repertoire of dynamics available to the LeuT-fold family, the most striking mechanistic differences between DraNramp and other structurally-studied LeuT-fold transporters concern the substrate transport routes. Most well-characterized members (including LeuT, BetP, and Mhp1) are $Na^+$-driven symporters of small organic molecules which have one or two $Na^+$-binding sites (*Perez and Ziegler, 2013*; *Rudnick, 2013*). Sodium binding at the highly-conserved Na2 site connects the 'bundle' (TM1) and 'scaffold' (TM8) domains while also shifting the conformational equilibrium to favor the outward-open state (*Claxton et al., 2010*; *Tavoulari et al., 2016*; *Zhao et al., 2011*). This Na2 site consists of hydroxyls from two consecutive serines/threonines on TM8 and four main-chain carbonyls (one from TM8, three from the unwound-region of TM1) (*Perez and Ziegler, 2013*; *Yamashita et al., 2005*). Intriguingly, the analogous location in DraNramp also contains highly-conserved hydroxyl-providing TM8 residues S327 (92%) and S328 (20% conserved, with another 74% as T), which may be remnants of the ancestral Na2 site conversion into a $H^+$ site in the Nramp clade. This hypothetical evolutionary switch has precedent within the LeuT-fold family, as the proton-coupled amino acid transporter ApcT analogously uses a conserved TM5 lysine (K158), whose sidechain protrudes into the Na2 location, as its primary proton-binding site (*Shaffer et al., 2009*). The sodium-to-proton switch may have evolved in Nramps to avoid simultaneously coordinating two metal cations ($Na^+$ coordination, like $Mn^{2+}$ coordination, requires ~6 oriented ligands, whereas $H^+$ binding requires a single sidechain).

LeuT and other bacterial homologs also antiport a proton as they return to an outward-open state (*Kantcheva et al., 2013*; *Zhao et al., 2010*; *Zomot et al., 2007*) via a conserved glutamate (E290) on TM7 (*Malinauskaite et al., 2016*), analogous to the highly-conserved N275 (100%) that lines DraNramp's intracellular vestibule. Available structures and MD simulations suggest that proton symport in ApcT and antiport in LeuT likely occur through the bulk opening and closing of the same permeation pathways used by the primary substrates (amino acids) (*Krishnamurthy and Gouaux, 2012*; *Malinauskaite et al., 2016*; *Shaffer et al., 2009*; *Shi and Weinstein, 2010*). In contrast DraNramp does $H^+$ uniport even when mutationally (G223W) or chemically (A61C-MTSET) precluded from opening the intracellular vestibule (*Figure 5*). We propose a proton route from D56 through D131 and into a conserved salt-bridge network between TMs 3, 4, 8, and 9 (*Figure 6*), which remain relatively stationary during the conformational change process (*Figure 2*). Indeed, evolutionary analysis reveals that this polar network is unique to the Nramp clade of the LeuT-family (*Cellier, 2016*); this region is mainly hydrophobic in both LeuT and ApcT (*Shaffer et al., 2009*; *Yamashita et al., 2005*). A parallel transport pathway for protons could alleviate the electrostatic problem of simultaneously stabilizing three added positive charges (the proton and divalent metal cation) in close proximity throughout a conformational change process, although other unrelated transporters are known to accommodate multiple positive charges within their binding sites during the transport cycle (*Vandenberg and Ryan, 2013*). The observed proton uniport in Nramp, requiring only subtle conformational rearrangements, is more reminiscent of $H^+$ shuttling in the CLC family of $Cl^-/H^+$ antiporters (*Accardi and Miller, 2004*; *Accardi and Picollo, 2010*; *Basilio et al., 2014*; *Miller, 2006*) than the canonical $Na^+$ transport seen in LeuT-family symporters.

It remains to be demonstrated whether the observed Nramp metal and proton transport truly constitute symport. The DraNramp proton and metal transport mechanism we outline, where primary and driving substrates enter via a common permeation pathway but exit via separate routes to the cytoplasm—with $H^+$ transfer perhaps triggering bulk conformational rearrangement needed for $Mn^{2+}$ release to occur if the substrates are in fact coupled—is thus far unique to the Nramp clade within the APC superfamily. This new model for Nramp transport therefore illustrates the evolutionary flexibility and adaptability of the shared LeuT fold.

## Materials and methods

**Key resources table**

| Reagent type (species) or resource | Designation | Source or reference | Identifiers |
|---|---|---|---|
| Gene (*Deinococcus radiodurans*) | DraNramp | Genomic DNA | Uniprot: Q9RTP8 |
| Strain (*Escherichia coli* C41(DE3)) | C41(DE3) | Lucigen | 60442–1 |
| Recombinant DNA reagent | pET21a | Novagen | 69740–3 |
| Antibody | Alexa 647-conjugated anti-His-tag | Qiagen | 35370 |
| Chemical compound | DDM | Anatrace | D310S |
| Chemical compound | DM | Anatrace | D322S |
| Chemical compound | LMNG | Anatrace | NG310 |
| Chemical compound | Monoolein | Anatrace | LCP18 |
| Chemical compound | Fura-2 | Life Technologies | F-1200 |
| Chemical compound | BCECF | VWR | 102987–506 |
| Chemical compound | POPE | Avanti Polar Lipids | 850757C |
| Chemical compound | POPG | Avanti Polar Lipids | 840457C |
| Chemical compound | Valinomycin | Sigma-Aldrich | V0627 |
| Chemical compound | 5K-PEG maleimide | Creative PEGWORKS | PLS-234 |
| Chemical compound | NEM | Sigma-Aldrich | 4260 |
| Chemical compound | MTSET | VWR | 89139–738 |
| Chemical compound | MTSEA | VWR | 89139–740 |
| Software, algorithm | HKL2000 | PMID: 27799103 | RRID:SCR_015547 |
| Software, algorithm | XDS | PMID: 20124692 | RRID:SCR_015652 |
| Software, algorithm | PHASER | PMID: 19461840 | RRID:SCR_014219 |
| Software, algorithm | PHENIX | PMID: 22505256 | RRID:SCR_014224 |
| Software, algorithm | coot | PMID: 20383002 | RRID:SCR_014222 |
| Software, algorithm | PyMOL | | RRID:SCR_000305 |
| Software, algorithm | PROPKA | PMID: 15215472 | |
| Software, algorithm | HMMER | PMID: 21593126 | RRID:SCR_005305 |
| Software, algorithm | MUSCLE | PMID: 19461840 | RRID:SCR_011812 |

*Continued on next page*

Continued

| Reagent type (species) or resource | Designation | Source or reference | Identifiers |
|---|---|---|---|
| Software, algorithm | ImageJ64 | | RRID:SCR_003070 |
| Software, algorithm | MATLAB | | RRID:SCR_001622 |

## Cloning of DraNramp

WT and mutant DraNramps were cloned in pET21-N8H as described (*Bozzi et al., 2016b*). All constructs were full-length, except the G223W crystallization construct was N-terminally truncated to residue 35; this deletion did not affect metal transport (*Bozzi et al., 2016b*). Mutations were made using the Quikchange mutagenesis protocol (Stratagene) and confirmed by DNA sequencing. Single-cysteine constructs also included the C382S mutation to remove the lone endogenous cysteine. The C41(DE3) *E. coli* strain was used for protein expression and in vivo assays.

## Expression and purification of DraNramp crystallography constructs

Six liters of DraNramp C41(DE3) cells were cultured as described (*Bozzi et al., 2016b*), pelleted and flash-frozen in liquid nitrogen. Proteins were purified at 4°C. Thawed cells were lysed by sonication in 40 mL load buffer (20 mM sodium phosphate, pH 7.5, 55 mM imidazole, 500 mM NaCl, 10% (v/v) glycerol) plus 1 mM PMSF, 1 mM benzamidine, and 0.3 mg/mL each DNAse I and lysozyme. Lysates were cleared for 20 min at 27,000 × $g$. Membranes were pelleted from supernatant at 230,000 × $g$ for 70 min, homogenized in 65 mL load buffer and flash-frozen in liquid nitrogen. Thawed membranes were solubilized for 1 hr, adding 1.5% (w/v) n-dodecyl-β-D-maltopyranoside (DDM), then spun at 140,000 × $g$ for 35 min to pellet debris. Pre-equilibrated Ni-sepharose beads (3 mL; GE Healthcare) were incubated with the supernatant for 1 hr, and washed with load buffer containing sequentially 0.03% DDM, 0.5% lauryl maltose neopentyl glycol (LMNG), and 0.1% LMNG. Protein was eluted in 20 mM sodium phosphate, pH 7.5, 450 mM imidazole, 500 mM NaCl, 10% (v/v) glycerol, 0.01% LMNG, concentrated to <0.5 mL in a 50 kDa cutoff centrifugal concentrator, and loaded onto a Superdex S200 10/300 (GE Healthcare) pre-equilibrated with SEC buffer (10 mM HEPES pH 7.5, 150 mM NaCl, 0.003% LMNG). Peak fractions were pooled, concentrated to ~20–40 mg/mL, aliquoted, and flash-frozen in liquid nitrogen.

## DraNramp crystallization, X-ray diffraction data collection, and structure determination

Protein (10–15 μL) was loaded into a 100 μL glass syringe attached to an LCP coupling device (Formulatrix). A second 100 μL syringe containing 1.5 volumes of liquid monoolein (T > 37°C) was attached to the coupling device, and the two solutions were mixed for 100 cycles using an NT8 (Formulatrix) in LCP mixing mode at 5 mm/s. LCP boluses (50–100 nL) and precipitant (600–1000 μL) were dispensed into 96-well LCP glass plates and incubated at room temperature (RT). Crystals of G223W (25 mg/mL) with 2.5 mM MnCl$_2$ in SEC buffer grown in 28% PEG400, 5 mM MnCl$_2$, 100 mM MES pH 6, 50 mM succinic acid pH 6, 10 mM spermidine pH 7, (10–30 μm square plates) were harvested after nine days using mesh loops (MiTeGen) to scoop the bolus, then flash-frozen in liquid nitrogen. Similar G223W crystals in the apo state were obtained after seven days with 26% PEG400, 100 mM MES pH 6, 50 mM succinic acid pH 6, 20 mM spermidine pH 7. Crystals of the analogous NEM-modified G223C/C382S ΔN34 (G223C retains close to WT-level metal transport before NEM-labeling (*Bozzi et al., 2016b*)) were obtained in the same condition. Crystals of G45R (22 mg/mL) grown in 20% PEGMME 550, 150 mM NaCl, 100 mM HEPES, pH 7.0 (~100 μm rods) were harvested after ~10 days.

Data were collected at Advanced Photon Source beamline 24ID-C. Crystals were located by grid scanning with a 70 μm beam at 70% transmission followed by focused grid scanning with a 10 μm beam at 100% transmission. Data wedges were typically collected from −30° to +30° in 0.2° increments using a 10 μm beam at 100% transmission. Two wedges from two crystals (G45R), 23 wedges from ~15 crystals (G223W with Mn$^{2+}$), or 39 wedges from ~20 crystals (G223W apo) were independently indexed and integrated then combined during scaling using HKL2000 (*Otwinowski and Minor, 1997*) to obtain complete datasets. Structures were determined using software provided by SBGrid (*Morin et al., 2013*). Initial

phases were obtained by molecular replacement in PHASER (*McCoy et al., 2007*) using our first DraN-ramp structure (PDB: 5KTE) as a search model for G45R and an in-progress G45R model for G223W. Model building and refinement were iterated in Coot (*Emsley and Cowtan, 2004*) and PHENIX (*Adams et al., 2010*), respectively. For all structures, positional and B-factor refinement with TLS restraints were used throughout, with torsion angle and NCS restraints for G45R, and secondary structure restraints for the Patch mutant. G45R contains two protein molecules in the asymmetric unit—chain A with residues 45–167 and 174–436 and chain B with residues 44–168 and 175–436 (RMSD 0.86 Å over 2899 atoms, 0.50 Å over all 386 Cαs)—and six fully or partly modeled monoolein molecules. Chain A was used for figures and analyses. The G223W $Mn^{2+}$-bound structure includes residues 39–436, ten full or partial monooleins, one spermidine molecule, and two $Mn^{2+}$ ions—one in the metal-binding site, one at a crystal-packing interface. The G223W apo structure includes residues 39–436 and six full or partial monooleins. The two G223W structures align with RMSD 1.41 Å over 3012 atoms, 1.08 Å over all 398 Cαs). The electron density for each TM is shown in *Figure 2—figure supplement 2A* for G45R and *Figure 2—figure supplement 2B* for G223W with $Mn^{2+}$, while *Figure 2—figure supplement 2C* shows the metal-binding site for both the $Mn^{2+}$-bound and apo G223W structures. The inward-open Patch mutant structure was updated to correct the position of intracellular loop 10–11 and the registry of TM11, and extend the N-termini of TMs 5, 7 and 9, the C-terminus of TM7, and extracellular loop 7–8, and improve the geometry of the Fab. The new model comprises residues 43–165, 170–236, 256–341, 351–436 of DraN-ramp, 1–129 and 132–213, and 1–213 of the Fab heavy and light chains, respectively, and three $Os^{3+}$ ions.

## In vivo metal transport assays

Metal uptake assays in *E. coli* were performed as described previously (*Bozzi et al., 2016a*). For each biological replicate reported in figure legends, a separate culture of transformed *E. coli* was grown and induced to express the exogenous Nramp construct.

## Purification of DraNramp constructs for proteoliposome assays

DraNramp constructs were cloned, expressed, and purified as described above, with the following changes: protein was purified from cell pellets in a single day, and washed/eluted from nickel beads in buffers with 0.03% DDM. Protein was concentrated to 2.5 mL and buffer-exchanged into 100 mM NaCl, 10 mM HEPES pH 7.5, 0.1% n-Decyl-β-D-Maltopyranoside (DM) on a PD10 desalting column. Protein concentrations were normalized to 1.2 mg/mL and aliquots were flash frozen in liquid nitrogen. Single-cysteine constructs A53C and A61C were purified in the presence of 1 mM DTT.

## Proteoliposome preparation

Adjusting the lipid composition (*Ehrnstorfer et al., 2017*) of a previous protocol (*Bozzi et al., 2016a*; *Tsai et al., 2014*), 75% w/w 1-palmitoyl-2-oleoyl-sn-glycero-3-phosphoethanolamine (POPE) was mixed with 25% w/w 1-palmitoyl-2-oleoyl-sn-glycero-3-phosphoglycerol (POPG) in chloroform (Avanti Polar Lipids), and then dried under nitrogen in a warm water bath, re-dissolved in pentane, and dried again. Lipids were resuspended at 20 mg/mL in 5 mM DM in KCl + NaCl/MOPS buffer (typically ~90 mM KCl, 30 mM NaCl, 0.5 mM or 10 mM MOPS pH 7). Protein was added at a 1:400 w/w ratio to lipid, and the mixture dialyzed at 4°C to remove the detergent in 10 kDa molecular weight cutoff dialysis cassettes against KCl + NaCl/MOPS buffer with 0.2 mM EDTA for 1 day, then with 0.1 mM EDTA for 1–3 days, then overnight at room temperature (RT) against KCl + NaCl/MOPS buffer. For A53C and A61C, 1 mM, and 0.5 mM DTT was included in the first two dialysis steps. Fluorescent dye (either 1:49 v/v 5 mM Fura-2 pentapotassium salt or 1:66 v/v 10 mM 2′,7′-bis(carboxyethyl)−5(6)-Carboxyfluorescein (BCECF) in dimethyl sulfoxide) was incorporated into proteoliposomes permeabilized by three freeze-thaw cycles in dry ice-ethanol and RT water baths (and sometimes stored at −80°C after the third freeze). Proteoliposomes were extruded through a 400 nM filter to create uniform-sized vesicles, buffer-exchanged 1–2 times on a PD10 desalting column pre-equilibrated with NaCl/ or KCl/10 mM MOPS pH 7 buffer. Peak proteoliposome-containing fractions were pooled to remove unincorporated dye.

## Proteoliposome transport assays and data analysis

Proteoliposomes loaded with either 100 μM Fura-2 or 150 μM BCECF were diluted into buffer containing appropriate [KCl] to establish the desired membrane potential (*Fitzgerald et al., 2017*; *Uzdavinys et al., 2017*) and aliquoted into 96 well black clear-bottom plates. Following baseline fluorescence measurement, 5X metal (750 μM final concentration unless otherwise noted) and valinomycin (100 nM final concentration) were added. Stocks of 100 mM $CdCl_2$, $MnCl_2$, and $Co(NO_3)_2$, as well as appropriate serial dilutions in water for concentration range experiments, were freshly diluted into appropriate NaCl or KCl buffer with pre-added valinomycin. To pre-modify cysteines (A53C or A61C when applicable), liposomes were diluted into buffer (120 mM NaCl, 10 mM MOPS pH 7) containing 3 mM MTSET, 3 mM MTSEA, or 4 mM NEM, and incubated at least 30 min at RT before beginning transport assays. Metal transport was monitored by measuring Fura-2 fluorescence at $\lambda_{ex}$ = 340 and 380 nm, at $\lambda_{em}$ = 510 nm. Proton transport was monitored by measuring BCECF fluorescence at $\lambda_{ex}$ = 450 and 490 nm, at $\lambda_{em}$ = 535 nm. To calculate concentrations of imported metal, the Fura-2 340/380 ratio and an experimentally determined $K_D$ value (*Hinkle et al., 1992*) was used for $Cd^{2+}$ as described previously (*Bozzi et al., 2016a*). For $Mn^{2+}$ and $Co^{2+}$, the fraction of Fura-2 340 and 380 fluorescence quenched, normalized to maximum observed quenching, was used to estimate imported metal. For proton uptake, the BCECF 450/490 ratio was used to calculate internal pH, which along with the known total internal buffer (0.5 mM) and dye (150 μM) concentration was used to calculate net proton import via the Henderson-Hasselbalch equation. The effect of divalent cations on BCECF fluorescence under analogous conditions to the liposome assay was tested (*Figure 5—figure supplement 3*), which showed that $Mn^{2+}$, $Co^{2+}$, $Zn^{2+}$, $Cd^{2+}$, $Ca^{2+}$ had no effect while $Fe^{2+}$ had a slight effect but of a much smaller magnitude than adding an equivalent concentration of $H^+$ or $OH^-$. Initial rates were calculated in Excel and Michaelis-Menten parameters were fit using MATLAB. For each technical replicate reported in figure legends, a separate aliquot of dye-loaded proteoliposomes was diluted into the appropriate outside buffer, including cysteine modifiers if applicable, then fluorescence time course data were collected before and after the addition of valinomycin, metal substrate, and/or ionomycin.

## Thrombin cleavage assay

DraNramp proteoliposomes with a 1:200 w/w ratio of protein to lipid were formed as described above, in 90 mM KCl, 30 mM NaCl, 10 mM MOPS pH 7, and extruded 19 times. Additional purified DraNramp was diluted to 0.1 mg/ml in the same buffer with 0.1% DM. A 1/16 vol of 250 mM Tris pH 8.25 was added to the proteoliposomes or detergent-solubilized protein to adjust the pH to 8.0 for optimal thrombin activity. After removing a 0 min aliquot, thrombin from human plasma (EMD Biosciences) was added to final concentrations of 2.5 or 10 U/mL, and timed aliquots were removed and quenched by adding excess PMSF (~3 mM) and sample buffer. Samples were run on SDS-PAGE and stained with Coomassie. Band intensities corresponding to the full-length (48.2 kDa) and thrombin-cleaved (42.9 kDa) proteins were calculated using ImageJ64, and the fraction of the protein in the lower band (corrected for the molecular weights) was determined.

## Cysteine accessibility measurements

Cells grown as for the uptake assay were washed once in labeling buffer (100 mM Tris pH 7.0, 60 mM NaCl, 10 mM KCl, 0.5 mM $MgCl_2$, 0.75 mM $CaCl_2$), resuspended at $OD_{600}$ = 2, and aliquoted 100 μL per well in a 96-well plate. A 1:1 NEM dilution series was prepared in labeling buffer at 8 mM; 100 μL of the appropriate 2X NEM solution was added to each well and incubated 15 min at RT. L-cysteine (10 μL of 200 mM) was added to quench NEM. Cells were washed twice in labeling buffer, pelleted, resuspended in 30 μL lysis and denaturing buffer (6 M urea, 0.1% SDS, 100 mM Tris pH 7) with 0.5 mM DTT and incubated 1 hr at 37°C. The lysate (10 μL) was mixed with 3.5 μL of 6 mM 5K-PEG maleimide (Creative PEGWORKS) in lysis and denaturing buffer, incubated 1 hr at 37°C, and the reaction terminated by adding sample buffer with β-mercaptoethanol. Protein was detected via SDS-PAGE and western blotting using an Alexa 647-conjugated anti-His-tag antibody (QIAGEN) and a Typhoon Imager (GE Healthcare). ImageJ64 was used to determine the % modification as described (*Bozzi et al., 2016b*). For each biological replicate reported in figure legends, a separate culture of transformed *E. coli* was grown and induced to express the exogenous Nramp construct.

## Sequence alignments

An alignment of 9289 Nramp sequences was obtained from a HMMER (*Finn et al., 2011*) search using the DraNramp sequence with an E-value of 1, filtered for sequences with just one domain, then filtered for sequences 400–600 residues long. Incomplete sequences and sequences lacking the canonical Nramp TM1 'DPGN' and TM6 'MPH' motifs were removed, yielding 6878 sequences. A seed of 92 diverse sequences were aligned using MUSCLE (*Edgar, 2004*), then HMMER was used with a gap threshold of 0.99 to create the final alignment (*Figure 2—source data 1*).

## Structural comparisons and analyses

Per-residue C$\alpha$ RMSD values were calculated using the ColorByRMSD PyMOL script, and whole-structure C$\alpha$ RMSD values using the PyMOL align command with cycles = 0. To generate the distance difference matrices, the pairwise distances between all C$\alpha$ atoms were calculated for each structure (inward, inward-occluded, and outward). Then, for each combination of two conformations, a distance difference matrix was calculated by taking the difference between the distance matrices corresponding to each conformation. These distance-difference values were then averaged for each pair of TM helices (after dividing TMs 1, 6, and 10 into 'a' and 'b' segments) using an RMSD-like calculation to obtain a 14×14 matrix for each pair of conformations. The python code used to perform these calculations and generate the resulting plots is available at GitHub: https://github.com/GaudetLab/coarse-grained-DDMP (copy archived at https://github.com/elifesciences-publications/coarse-grained-DDMP). p$K_a$ values were calculated using PROPKA (*Dolinsky et al., 2004*) with CHARMM forcefields.

## Data availability

The accession number for the DraNramp crystal structures reported in this paper are G45R inward-occluded, PDB ID: 6C3I; G223W ΔN34 outward-open with $Mn^{2+}$, PDB ID: 6BU5; G223W ΔN34 outward-open apo, PDB ID: 6D91; revised inward open, PDB ID: 6D9W. The unprocessed diffraction images were deposited in the SBGrid Data Bank (https://data.sbgrid.org/) with SBGDB ID: 567 (G45R); 564 and 576 (G223W ΔN34 $Mn^{2+}$-bound and apo respectively); and 332, 333, and 334 (inward-open). The raw biochemical data that support the findings of this study are available from the corresponding author upon reasonable request.

## Acknowledgements

We would like to thank Andrew Kruse for assistance with LCP crystallography, Anthony Hesser for initial DraNramp crystallization, and Chris Miller, Joe Mindell, Niels Bradshaw and Gaudet lab members for helpful discussions. This work was funded in part by NIH grant 1R01GM120996 (to RG) and a Jane Coffin Childs Postdoctoral Fellowship (to CMZ). We thank the NE-CAT beamline staff at the Advanced Photon Source (Argonne, IL, USA) for help with data collection. NE-CAT is funded by NIH (P41 GM103403 and S10 RR029205), and the Advanced Photon Source by the US Department of Energy (DE-AC02-06CH11357).

## Additional information

### Funding

| Funder | Grant reference number | Author |
| --- | --- | --- |
| National Institute of General Medical Sciences | 1R01GM120996 | Rachelle Gaudet |
| Jane Coffin Childs Memorial Fund for Medical Research | | Christina M Zimanyi |

The funders had no role in study design, data collection and interpretation, or the decision to submit the work for publication.

## Author contributions
Aaron T Bozzi, Conceptualization, Resources, Data curation, Formal analysis, Validation, Investigation, Visualization, Methodology, Writing—original draft, Writing—review and editing, Performed all functional assays and analyzed the resulting data, Obtained crystals and diffraction data, Solved the structures of G223W ΔN34 DraNramp, with assistance at each stage from JMN, CMZ, and/or RG, Guided BKL as he designed, cloned, and initially validated the panel of tryptophan mutants; Christina M Zimanyi, Conceptualization, Data curation, Formal analysis, Validation, Investigation, Visualization, Methodology, Writing—original draft, Writing—review and editing, Obtained crystals and diffraction data and solved the structure of G45R DraNramp, Assisted ATB with the structures of G223W ΔN34 DraNramp; John M Nicoludis, Conceptualization, Investigation, Methodology, Writing—review and editing, Assisted ATB with the structures of G223W ΔN34 DraNramp; Brandon K Lee, Formal analysis, Investigation, Methodology, Writing—review and editing, Designed, cloned, and initially validated the panel of tryptophan mutants with ATB; Casey H Zhang, Data curation, Software, Investigation, Methodology, Writing—review and editing, Computationally analyzed Nramp sequences and structures using distance difference matrices; Rachelle Gaudet, Conceptualization, Resources, Supervision, Funding acquisition, Validation, Visualization, Project administration, Writing—review and editing, Corrected the registry and re-refined the original DraNramp structure, Assisted ATB with the structures of G223W ΔN34 DraNramp

## Author ORCIDs
Aaron T Bozzi  https://orcid.org/0000-0003-4245-8725
Christina M Zimanyi  https://orcid.org/0000-0002-6782-507X
John M Nicoludis  https://orcid.org/0000-0002-3755-7844
Rachelle Gaudet  https://orcid.org/0000-0002-9177-054X

## Decision letter and Author response
Decision letter https://doi.org/10.7554/eLife.41124.043
Author response https://doi.org/10.7554/eLife.41124.044

# Additional files

## Supplementary files
• Transparent reporting form
DOI: https://doi.org/10.7554/eLife.41124.021

## Data availability
Source data for the Nramp sequence alignment have been provided as Figure 2-source data 1. Structural coordinates and structure factors for each crystal structure have been deposited in the PDB under accession codes 6D9W, 6C3I, 6BU5, and 6D91. The unprocessed X-ray diffraction images have been deposited in the SBGrid Data Bank under accession codes 332, 333, 334, 567, 564, and 576. All other data generated or analyzed in this study are included in the manuscript.

The following datasets were generated:

| Author(s) | Year | Dataset title | Dataset URL | Database and Identifier |
|---|---|---|---|---|
| Zimanyi CM, Bozzi AT, Gaudet R | 2018 | Crystal structure of the Deinococcus radiodurans Nramp/MntH divalent transition metal transporter G45R mutant in an inward occluded state | https://www.rcsb.org/structure/6C3I | Protein Data Bank, 6C3I |
| Bozzi AT, Zimanyi CM, Nicoludis JM, Gaudet R | 2017 | Crystal structure of the Deinococcus radiodurans Nramp/MntH divalent transition metal transporter in the outward-open, manganese-bound conformation | https://www.rcsb.org/structure/6BU5 | Protein Data Bank, 6BU5 |
| Bozzi AT, Zimanyi CM, Nicoludis JM, Gaudet R | 2018 | Crystal structure of the Deinococcus radiodurans Nramp/MntH divalent transition metal transporter in the outward-open, apo conformation | https://www.rcsb.org/structure/6D91 | Protein Data Bank, 6D91 |

| Gaudet R, Bane LB, Weihofen WA, Singharoy A, Zimanyi CM, Bozzi AT | 2018 | Crystal structure of Deinococcus radiodurans MntH, an Nramp-family transition metal transporter, in the inward-open apo state | https://www.rcsb.org/structure/6D9W | Protein Data Bank, 6D9W |
| Zimanyi CM, Bozzi AT, Gaudet R | 2016 | X-Ray Diffraction data from Deinococcus radiodurans Nramp/MntH divalent transition metal transporter mutant G45R, source of 6C3I structure | https://dx.doi.org/10.15785/SBGRID/567 | SBGrid Data Bank, 10.15785/SBGRID/567 |
| Bozzi AT, Zimanyi CM, Nicoludis JM, Gaudet R | 2017 | X-Ray Diffraction data from Deinococcus radiodurans Nramp/MntH divalent transition metal transporter in the outward-open, manganese-bound state, source of 6BU5 structure | https://dx.doi.org/10.15785/SBGRID/564 | SBGrid Data Bank, 10.15785/SBGRID/564 |
| Bozzi AT, Zimanyi CM, Nicoludis JM, Gaudet R | 2017 | X-Ray Diffraction data from Deinococcus radiodurans Nramp/MntH divalent transition metal transporter mutant G223W, apo, source of 6D91 structure | https://dx.doi.org/10.15785/SBGRID/576 | SBGrid Data Bank, 10.15785/SBGRID/576 |

The following previously published datasets were used:

| Author(s) | Year | Dataset title | Dataset URL | Database and Identifier |
| --- | --- | --- | --- | --- |
| Bane LB | 2010 | X-Ray Diffraction data from Deinococcus radiodurans MntH in complex with Fab, source of 5KTE structure | https://dx.doi.org/10.15785/SBGRID/332 | SBGrid Data Bank, 10.15785/SBGRID/332 |
| Bane LB, Weihofen WA, Gaudet R | 2010 | X-Ray Diffraction data from Deinococcus radiodurans MntH in complex with Fab, source of 5KTE structure | https://dx.doi.org/10.15785/SBGRID/333 | SBGrid Data Bank, 10.15785/SBGRID/333 |
| Bane LB, Weihofen WA, Gaudet R | 2011 | X-Ray Diffraction data from Deinococcus radiodurans MntH in complex with Fab, source of 5KTE structure | https://dx.doi.org/10.15785/SBGRID/334 | SBGrid Data Bank, 10.15785/SBGRID/334 |

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
