## [Decision Letter]

Thank you for submitting your article "Structures in multiple conformations reveal distinct transition metal and proton pathways in an Nramp transporter" for consideration by *eLife*. Your article has been reviewed by three peer reviewers, and the evaluation has been overseen by José Faraldo-Gómez as Reviewing Editor and Michael Marletta as the Senior Editor. All reviewers have agreed to reveal their identity: Ming Zhou (Reviewer #1); Dirk Slotboom (Reviewer #2); Gary Rudnick (Reviewer #3).

The reviewers have discussed the reviews with one another and with the Reviewing Editor, who has drafted this decision to help you prepare a revised submission.

The reviewers have no major concerns with your structural work, which they see as an important advancement in the field, and in this regard they recommend only minor changes to clarify the presentation and discussion of the data, as noted below. The reviewers are also very impressed by the effort invested in the functional characterization of the transporter. However, they also find that the current interpretation of this data in mechanistic terms is less than compelling, and that a clear conclusion would require additional functional assays and some key controls, which are also enumerated below. Should you be unable to provide this new data, we ask that you thoroughly reformulate the discussion of your results, so as to more accurately convey the significance of the existing data, and the questions that remain unanswered; in this case, the conclusions, Abstract and title ought to be amended accordingly.

A central concern is whether the observation of uncoupled proton fluxes under specific experimental conditions and only for the outward-facing form of the transporter is actually informing on the nature of the mechanism of the transporter as it carries out its presumed biological function, i.e. symport. While your data appears to enable you to delineate the pathway for passive proton leak, the claim that this process coincides with that of proton binding and release in the context of active proton-coupled metal uptake does not seem to be directly supported by the data presented. That the measured proton uptake increases in the presence of a transportable substrate does not fully resolve this question, as it is unclear whether this increase is due to more of the uncoupled proton flux, to substrate-stimulated flux, to coupled proton-substrate uptake, or to some combination of these processes. Indeed, there are examples of ion fluxes through transporters that are stimulated by substrate binding or transport (EAATs and biogenic amine NSS transporters), where those fluxes are mechanistically independent from substrate transport.

A related key concern pertains to the molecular mechanism you outline in Figure 7C, in that it does not appear to explain symport, except in very specific conditions. In this model, 1:1 symport appears to require (a) that protons bind to the outward-facing transporter (only) from the outside, even though you also propose that a pathway exists to the intracellular side; (b) that protons are ejected upon metal binding (only) to the inside, even though the protein is outward-facing; and (c) that after metal release into the interior, the apo outward-facing transporter does not (ever) mediate uncoupled outward proton flux, even though this flux occurs in metal-free conditions. In this model, therefore, the directionality of the symport reaction appears to require the continued presence of inward electrochemical gradients for both metal and proton (or that the intrinsic microscopic reversibility of each of these steps is violated). By definition, however, a symport mechanism must explain co-transport even when one of the transported substrates is subject to an opposing driving force, or none at all.

Essential revisions

1) In order to preserve the current claims in regard to the mechanism of proton-driven symport, and its relationship to the observation of uncoupled fluxes, it would be essential to (a) demonstrate the actual symport activity of the transporter; and (b) clarify how much (if any) of the measured proton flux is actually coupled to active metal transport. Direct information on the flux stoichiometry (how many protons are transported for each transported metal ion) and the reversal potential for the proton fluxes would help to support to clarify these questions. Demonstrating that a gradient of Mn^2+^ could drive protons across the membrane would be a powerful argument in favor of symport, provided there were no other driving force for proton influx. In all the experiments presented, a K^+^ diffusion potential creates an electrical driving force for proton influx. If this potential was neutralized or reversed, would protons be driven into or out of the proteoliposomes by a transmembrane gradient of metal ion substrate? Our view is that in the absence of these key experiments, it would be premature to make any statements about the mechanism that (hypothetically) couples the translocation of protons and metals through the membrane, and how it might be similar or different from that of proton leak.

2) The interpretation of the experiments described in subsection “DraNramp proton transport does not require large conformational change”, in which membrane permeable and impermeable reagents were used to modify cysteines, relies heavily on the assumption that the protein reconstitutes in both orientations with similar probability. This assumption must be tested experimentally. Some proteins, like LacY, have a strong preference for insertion in one orientation. Without direct experimental evidence in this regard, not based on transport measurements, alternative interpretations should be put forward that consider the case that the protein has a strong preference for insertion in one orientation, like other transporters e.g. LacY.

3) Finally, the authors must elaborate on what is the expected response of the proton indicator to the presence of divalent cations, ideally with a control experiment or a reference to existing data in this regard.

---

## [Author Response]

The reviewers have no major concerns with your structural work, which they see as an important advancement in the field, and in this regard they recommend only minor changes to clarify the presentation and discussion of the data, as noted below. The reviewers are also very impressed by the effort invested in the functional characterization of the transporter. However, they also find that the current interpretation of this data in mechanistic terms is less than compelling, and that a clear conclusion would require additional functional assays and some key controls, which are also enumerated below. Should you be unable to provide this new data, we ask that you thoroughly reformulate the discussion of your results, so as to more accurately convey the significance of the existing data, and the questions that remain unanswered; in this case, the conclusions, Abstract and title ought to be amended accordingly.A central concern is whether the observation of uncoupled proton fluxes under specific experimental conditions and only for the outward-facing form of the transporter is actually informing on the nature of the mechanism of the transporter as it carries out its presumed biological function, i.e. symport. While your data appears to enable you to delineate the pathway for passive proton leak, the claim that this process coincides with that of proton binding and release in the context of active proton-coupled metal uptake does not seem to be directly supported by the data presented. That the measured proton uptake increases in the presence of a transportable substrate does not fully resolve this question, as it is unclear whether this increase is due to more of the uncoupled proton flux, to substrate-stimulated flux, to coupled proton-substrate uptake, or to some combination of these processes. Indeed, there are examples of ion fluxes through transporters that are stimulated by substrate binding or transport (EAATs and biogenic amine NSS transporters), where those fluxes are mechanistically independent from substrate transport.A related key concern pertains to the molecular mechanism you outline in Figure 7C, in that it does not appear to explain symport, except in very specific conditions. In this model, 1:1 symport appears to require (a) that protons bind to the outward-facing transporter (only) from the outside, even though you also propose that a pathway exists to the intracellular side; (b) that protons are ejected upon metal binding (only) to the inside, even though the protein is outward-facing; and (c) that after metal release into the interior, the apo outward-facing transporter does not (ever) mediate uncoupled outward proton flux, even though this flux occurs in metal-free conditions. In this model, therefore, the directionality of the symport reaction appears to require the continued presence of inward electrochemical gradients for both metal and proton (or that the intrinsic microscopic reversibility of each of these steps is violated). By definition, however, a symport mechanism must explain co-transport even when one of the transported substrates is subject to an opposing driving force, or none at all.

We are grateful for the thorough and insightful reviews. This revised manuscript addresses the comments in the decision letter, as detailed on the following pages below. We believe our revisions have significantly improved the quality of the manuscript.

We have edited the model we present in Figure 7C to remove the word “symport” which we agree we have not conclusively demonstrated for DraNramp (or any Nramp homolog), with “metal and proton transport,” to describe the observed metal transport and metal-stimulated proton transport we measure in Figure 5. We also now stress that this model is meant to show the dominant activity of the protein in its physiological context in which prevailing pH gradients and voltage would favor outward-to-inward transport. In the figure legend we now call this the DraNramp “model under prevailing physiological conditions (negative inside membrane potential, acidic-outside pH gradient)” and in the figure we indicate the presence of these favorable electrochemical gradients.

Essential revisions1) In order to preserve the current claims in regard to the mechanism of proton-driven symport, and its relationship to the observation of uncoupled fluxes, it would be essential to (a) demonstrate the actual symport activity of the transporter; and (b) clarify how much (if any) of the measured proton flux is actually coupled to active metal transport. Direct information on the flux stoichiometry (how many protons are transported for each transported metal ion) and the reversal potential for the proton fluxes would help to support to clarify these questions.

Our aim in this paper is to present the structural basis for DraNramp proton and metal transport without delving in-depth into the coupling mechanism. While we can now readily perform kinetic analyses of metal and proton fluxes, we concur with the reviewers that a formal demonstration of coupled symport would require experiments assessing transport under different thermodynamic equilibria conditions, which will require significant modification and optimization of our in vitro system and are beyond the scope of this study.

We therefore carefully modified our manuscript to make clear that we have not conclusively demonstrated a symport mechanism in DraNramp, and to more generally stress the limits of current knowledge. These changes include minimizing the use of terms such as “symport”, “co-transport”, and “coupling”. We also added a statement in the Introduction explaining that thermodynamic coupling has not yet been demonstrated in Nramps, because to our knowledge no study in the literature has met this standard with any Nramp transporter. Nevertheless, many Nramp papers starting with the original characterization (Gunshin et al., 1997) have included significant speculation and discussion of a potential symport mechanism based on data similar to what we present. We thus consider it important and appropriate to discuss the possibility of symport in the current work, while remaining open to other possibilities regarding the transport mechanism and modified the discussion to make explicit these possibilities.

Demonstrating that a gradient of Mn^2+^ could drive protons across the membrane would be a powerful argument in favor of symport, provided there were no other driving force for proton influx. In all the experiments presented, a K^+^ diffusion potential creates an electrical driving force for proton influx. If this potential was neutralized or reversed, would protons be driven into or out of the proteoliposomes by a transmembrane gradient of metal ion substrate? Our view is that in the absence of these key experiments, it would be premature to make any statements about the mechanism that (hypothetically) couples the translocation of protons and metals through the membrane, and how it might be similar or different from that of proton leak.

As we showed in a forthcoming manuscript (Bozzi et al., 2018, currently deposited on bioRxiv), Mn^2+^ transport is highly dependent on voltage under our currently established experimental conditions, such that the WT protein does not transport metal in the absence of a favorable ΔΨ, which unfortunately precludes us from testing whether a metal gradient could drive proton influx at ΔΨ = 0 or ΔΨ > 0. Therefore, while we are very much interested in determining whether there is indeed thermodynamic coupling between proton and metal transport, it will require exploring and optimizing our experimental system beyond the scope of this current structure-focused manuscript.

The mutagenesis data in the forthcoming manuscript mentioned above (Figure 6C and Figure 6—figure supplement 2) showed that neutralizing the same four residues that eliminated voltage-driven H^+^ uniport (D56, D131, E134, and H232) also eliminated (D56, D131, H232) or drastically-reduced (E134) metal-stimulated H^+^ transport. These results support our assertion that the metal-stimulated H^+^ transport likely follows the same transport pathway as the H^+^ uniport. We therefore now state in the discussion: “For metal-stimulated proton transport, Mn^2+^ binding likely stimulates proton transfer into the same salt-bridge network, perhaps by directly ejecting a proton from D56 in the metal-binding site. Indeed, in a separate study we show that neutralizing mutations to the same four residues that eliminated H^+^ uniport (Figure 6E) also eliminated (D56, D131, H232) or severely-reduced (E134) H^+^ fluxes in the presence of added Mn^2+^, with mutants to D131, E134, and H232 retaining significant metal-transport ability despite a lack of proton transport (Bozzi et al., bioRxiv, 2018). However, the precise order of events for proton-metal transport, including whether it is indeed a co-transport mechanism, remains undetermined at this time, and additional transport mechanisms are possible.”

2) The interpretation of the experiments described in subsection “DraNramp proton transport does not require large conformational change”, in which membrane permeable and impermeable reagents were used to modify cysteines, relies heavily on the assumption that the protein reconstitutes in both orientations with similar probability. This assumption must be tested experimentally. Some proteins, like LacY, have a strong preference for insertion in one orientation. Without direct experimental evidence in this regard, not based on transport measurements, alternative interpretations should be put forward that consider the case that the protein has a strong preference for insertion in one orientation, like other transporters e.g. LacY.

We agree that determining the liposome orientation profile of DraNramp is necessary to fully interpret our liposome assay data. We therefore adapted an assay used in other transporters (Tsai et al., 2012 and Tsai et al., 2013) that uses thrombin to cleave DraNramp at a naturally-occurring cleavage site in the non-conserved, disordered N-terminal region before TM1a. In a new figure (Figure 5—figure supplement 2) we show that while thrombin fully cleaves DraNramp in detergent, only ~50% of liposome-reconstituted DraNramp is cleaved by added external thrombin. This result indicates that ~50% of the protein reconstitutes in an inside-out orientation with an exposed N-terminal, while the remaining ~50% reconstitutes in an outside-out orientation with its N-terminal enclosed in the proteoliposome and thus protected from external thrombin. We believe this experiment convincingly demonstrates that DraNramp indeed orients randomly in our proteoliposome production protocol.

We added the relevant information to the Materials and methods section and describe these results at the end of the “DraNramp proton transport does not require large conformational change” section:

“To confirm that a mix of inside-out and outside-out transporters were indeed present in proteoliposomes, we assessed the susceptibility of DraNramp to thrombin cleavage at a naturally-occurring site (Gallwitz et al., 2012) in the protein’s non-conserved, disordered N-terminal region (Figure 5—figure supplement 2A-C). While thrombin fully cleaved DraNramp in detergent, in proteoliposomes the cleaved protein population plateaued at ~50% (Figure 5—figure supplement 2D-F). This cleaved portion likely corresponds to inside-out oriented protein with an exposed N-terminal region, with the remaining ~50% of protein therefore outside-out oriented with the N-terminus inside the liposome and thus protected from thrombin cleavage (Tsai et al., 2012; Tsai et al., 2013).”

3) Finally, the authors must elaborate on what is the expected response of the proton indicator to the presence of divalent cations, ideally with a control experiment or a reference to existing data in this regard.

We performed the requested control experiment, included as the new Figure 5—figure supplement 3. Mn^2+^, Co^2+^, Zn^2+^, Cd^2+^, and Ca^2+^ have no significant effect on BCECF fluorescence, while Fe^2+^ had a slight effect, only ~12% in magnitude than the equivalent H^+^ concentration. Thus, the presence of divalents should not be a concern in all the assays presented in this manuscript, and this control experiment supports our conclusions based on these data.